# Strategic land reallocation enhances carbon sequestration and biodiversity protection without compromising agricultural productivity in Great Britain
Sarah S. Gall [✉], Tom Harwood, Michael Obersteiner & Jim W. Hall

Due to the negative environmental consequences of current land-use, and land's important role regarding carbon, biodiversity and food security, there is an urgent interest in reforming land-use. Policy objectives for tree planting to sequester carbon and the protection of land to increase biodiversity require land reallocation, which leads to inevitable trade-offs. Here, we evaluate the trade-offs between three objectives for rural land: agricultural/forestry production, carbon sequestration and biodiversity, by calculating metrics for these three objectives on a 500 m grid covering Great Britain. We use a multi-objective optimisation that allows us to explore the full option space of possible land conversions and identify the land allocations that entail limited trade-offs. Our results show that current land-use in Great Britain is far from optimal for any combination of objectives. We identify the locations where carbon sequestration and biodiversity can be substantially improved without compromising overall agricultural production, provided conversions are located carefully.

Climate change and biodiversity loss are two of the biggest ecological problems of our time[1]. Both challenges are closely linked to how land is used[2-4]. Currently, 70% of the global land mass is under human use[4]. This anthropogenic appropriation of land has had dramatic effects on the natural environment, with agriculture being the leading cause of global biodiversity loss[5,6] and a major source of greenhouse gas (GHG) emissions[7]. There are now worldwide political ambitions to confront both challenges, ratified in the Paris Agreement and the Kunming-Montreal Global Biodiversity Framework. To meet the objectives defined in these frameworks, an urgent need for more sustainable land-use has been identified[4,8]. This includes sustainable land-management practices and sufficient areas of land allocated for biodiversity and climate mitigation[5,9]. At the same time, it is crucial to consider the interactions between biodiversity, land-based climate mitigation, and food production, and manage trade-offs between these objectives[10]. This combined challenge of reversing biodiversity and avoiding climate change while providing enough food has been described as the 'triple challenge' by the World Wide Fund For Nature (WWF)[11]. To tackle this challenge, new approaches combining food security, ecosystem services, and biodiversity, as well as climate mitigation and adaptation, need to be considered when developing new land-use strategies[12-15]. Due to the heterogeneous character of landscapes, there is not one universally valid solution for sustainable land-use decisions, but instead, context-specific, integrated and multi-dimensional transformation strategies must be developed[16].

Like many other countries, the United Kingdom (UK) has acknowledged the need to transition to more sustainable land-use. The government has set a vision for environmental conservation[17] aiming "to be the first generation to leave the natural environment of England in a better state than it inherited"[18] and has also set the goal to achieve net zero GHG emissions by 2050[19]. To achieve this goal, there is strong emphasis upon afforestation[20] (the Climate Change Committee (CCC) suggests at least 17% woodland cover by 2050[21]; the Woodland Trust suggests up to 19% woodland cover to achieve carbon neutrality by 2050[22]) and sustainable land management as well as rethinking the livestock production sector[23,24]. Currently, the land-use sector is responsible for about 12% of total UK GHG emissions[21]. For biodiversity, a range of different objectives has been defined by the UK and its devolved nations, including the creation and restoration of substantial new areas of wildlife-rich habitats[25,26] and "reversing biodiversity loss by 2030", as stated in the Leader's Pledge for Nature and formalised in the Environment Act 2021. At the same time, there is an increasing emphasis on domestic food production and self-sufficiency[27], recognising climate and other risks to agricultural production globally and hence the insecurity of food imports, which account for 46% of consumption[28].

Environmental Change Institute, University of Oxford, Oxford, UK. [✉]e-mail: sarah.gall@spc.ox.ac.uk

This coincides with the UK leaving the European Union's (EU) Common Agricultural Policy, which has a substantial effect on regulations and agricultural subsidies and, therefore, farms in the UK. The Agricultural Act 2020 and the Environmental Land Management scheme provide a legislative framework to replace the direct payments scheme under the EU's Common Agricultural Policy with a payments for public goods scheme[29]. It forms the basis for phasing out direct payments over an agricultural transition period of seven years until the end of 2027[30]. Following these changes in agricultural subsidies, big shifts can be expected in the UK's agricultural sector. In particular, some smaller livestock farms may go out of business[31], leaving behind numerous areas that could be converted economically and ecologically beneficially to other land-use types. Those shifts entail the need to reorganise land and consider the most efficient and beneficial land uses. This provides a unique opportunity for policymakers to rethink land-use policy in the UK and to design and implement new environmental land management instruments to achieve environmental objectives like climate mitigation and adaptation[32]. Making sure that new policies and land-use strategies contribute to achieving climate and biodiversity objectives without threatening food production is of high interest. At the same time, the evidence needed to guide new land-use policies is missing[33].

Land-use modelling can help deliver the evidence needed to design new strategies and policies. There is a range of existing models and approaches for modelling land-use, with each type of model being suitable for answering a certain set of policy-relevant questions. A detailed overview of existing land-use models and the policy-relevant questions they can answer can be found in Supplementary Table 1. A common approach is the use of calculators based on exogenously specified land-use scenarios, which evaluate the consequences of those scenarios with respect to different metrics, such as the FABLE calculator[34] or the CCC land-use scenarios[21,32]. These models give essential insights for land-use target setting but are usually not spatially explicit and do not allow the exploration of spatial trade-offs between different land-use objectives. Additionally, they consider a few predefined scenarios and, therefore, cannot explore the full range of possibilities of future land-use patterns.

Another common approach for analysing land-use change is the use of agent-based models[35–37], which consider different groups of agents and their interactions with each other and with external drivers. While this approach can give an interesting insight into how agents might react to new policy interventions, it does not allow us to identify the land conversions that may be desirable in the first place.

Integrated land-use models, which are another common tool in land-use modelling, combine economic and environmental aspects and are often spatially explicit[38]. Examples are GLOBIOM, which is an integrated global model of land-use competition between agriculture, timber production and energy crops[39], or MagPIE, a global land-use allocation model based on economic conditions and land and water availability[40]. The downside of these integrated models is that they usually have a very coarse spatial resolution and consider aggregated economic regions rather than individual countries[38], making them less suitable for national policy-making and simulating landscape-scale features. Additionally, many of the large-scale integrated agricultural and land-use models do not incorporate effects on local biodiversity, even though some have published biodiversity scenarios as contributions to the 'bending the curve' discussion[41]. Another integrated model that considers many different sectors, including biodiversity, is the NEVO tool (Natural Environment Valuation Online tool), which optimises land allocation based on the market value of the considered ecosystem services[42]. By assigning a monetary value to all considered benefits (including water quality, recreational purposes, biodiversity, etc.), it implicitly assigns weightings that drive the optimisation outcome.

While all these models help answer important aspects related to the design of new land-use policies, none of them is suitable for evaluating the land-use interdependencies between different objectives from an explicitly spatial perspective. For this purpose, spatial trade-off modelling is more appropriate. This type of modelling is used in many contexts to identify optimal locations for specific land uses and spatial trade-offs. A typical application considers ecosystem service trade-offs[43–46], though methodologies vary and are usually applied in a particular context on a regional or local scale. The focus is generally more on the local interactions between the services than on evaluating the trade-offs for policymaking on a national scale.

One of these studies explored trade-offs between agriculture and ecosystem services under different agricultural management trajectories for Europe[47]. Similar studies have been conducted on a global scale for trade-offs between food, water and carbon[48]; carbon storage, biodiversity, water use, and food supply[49]; or biodiversity, carbon, and water[50], which are insightful. However, spatially explicit country-level analysis is still needed to provide more insights for national policy measures and actual decision-making.

There is a lack of studies that address the triple challenge of carbon sequestration, food (and timber) production, and biodiversity on limited land for national policy making. Additionally, existing models do not provide a spatial overview of trade-offs for national policymaking that includes the full range of land-use possibilities without limiting the outputs by only exploring a handful of scenarios or influencing the outcomes with predefined weightings. Therefore, we present a spatially explicit approach targeted at decision-makers in Great Britain (GB) that shows the entire decision space for the full range of potential priorities while pointing out synergies and trade-offs that need to be considered.

In this paper, we assess the full range of land-use trade-offs and synergies along the dimensions of carbon sequestration, biodiversity, and (food and timber) production at high geographic resolution. This allows us to identify the most beneficial place-specific land-use choices for achieving those objectives. Our analysis is implemented on a $500 \times 500$ m grid covering GB, entailing 814,004 grid cells. For each grid cell and each of the four land-use categories (arable, pasture, plantation forest, and semi-natural habitat), we quantify the potential location-specific benefits for carbon sequestration, production, and biodiversity of maintaining a land-use or converting the land to each of the other land-use categories (see Methods). By analysing the full range of possible land-use conversions in the country, we enable decision-makers to explore the entire options space as a basis for designing new targeted land-use policies. Furthermore, we recognise that considerable land-use changes (as a percentage of the overall national land area) are politically challenging to implement. To this end, we explore a range of land-use change budgets (i.e., the total area over which land-use changes can take place), providing stakeholders with the flexibility to explore the implications of different percentages of land-use change. Finally, we identify the most robust land conversions to changing priority weightings.

## Results
### Pareto-optimal land-uses
For each grid cell, we identified the land-use conversions that maximise and minimise each of the three objectives: Carbon sequestration, (food and timber) production, and biodiversity. The performance of a land allocation scenario in terms of carbon sequestration includes carbon fluxes from land conversion, emissions from agriculture, and carbon sequestration by vegetation and forest growth measured in million t $CO_2$-eq·yr$^{-1}$. Production includes the generated output from agriculture and timber production and is measured in billion £·yr$^{-1}$. Biodiversity is quantified with a unitless indicator that combines species occurrence probabilities and habitat condition values and is normalised between 0 and 1. Aggregating these maximum and minimum performances over the entire country gives the maximum and minimum potential outcomes for unlimited land conversion (see Table 1). Henceforth, we normalise this range, allocating a performance of 0 to the minimum and 1 to the maximum for each of the three objectives.

Next, we examine all the possible trade-offs between the three objectives. We discretise the continuous weighting combinations into a step size of five, where 100%-0%-0% would fully prioritise the first objective while 35%-35%-30% would consider all three objectives nearly equally, resulting in 231 vectors of 3-way weightings for each cell. The step size was chosen based on preliminary tests to balance the number of scenarios and the

**Table 1 | Performance if each of the objectives was minimised and maximised, with no limits on the area of land-use changes**

| | Carbon sequestration (million t $CO_2$-eq·yr$^{-1}$) | Food and forestry production (billion £·yr$^{-1}$) | Biodiversity (species occurrence based indicator, see Methods) |
|---|---|---|---|
| Minimum performance across all possible land-use changes (normalised performance of 0) | −143.39 | 0.062 | 159,884 |
| Maximum performance across all possible land use changes (normalised performance of 1) | 98.25 | 25.42 | 349,740 |

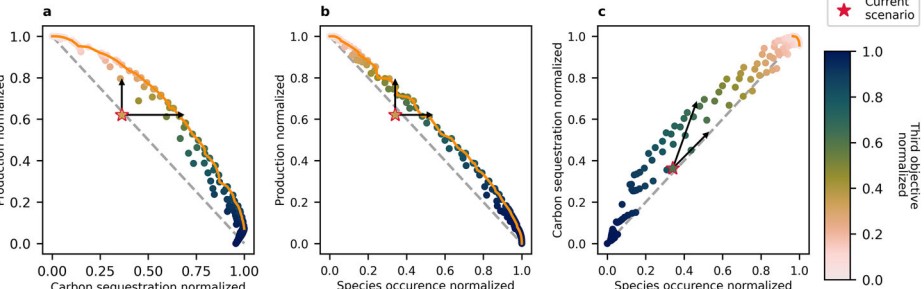

**Fig. 1 | Two-dimensional pairwise Pareto-efficient scenarios and the resulting Pareto frontiers.** Pareto-efficient points with their performance in terms of **a** carbon sequestration and production, **b** biodiversity and production, and **c** biodiversity and carbon sequestration, with the normalised performance of the third objective is shown with a colour gradient. The frontier of pairwise Pareto-efficient scenarios, when considering only the combination of two of the objectives, is shown in orange, and the performance of the current state is indicated by the red star. The arrows visualise the distance from the current scenario to the Pareto frontier along the three axes, and the light grey dashed line shows the linear connection between the objectives for comparison.

resolution needed to create a smooth Pareto frontier (see Supplementary Figs. 1–4). For each of these combinations of objectives, the best land-use for each cell is identified, and the corresponding benefit is recorded for the three objective metrics. This enables us to identify the optimal land-uses for every grid cell in the country for any combination of objectives and to plot the Pareto frontier of possible performance (Fig. 1). Additionally, it is worth noting that there is only an indirect connection between the normalised benefit results (normalised between 0 and 1) and the objective weightings (expressed as percentages). While an objective weighting of 100%-0%-0% leads to the maximum outcome for the first objective, it does not necessarily result in a minimum for the other two objectives, as there may be synergies between them.

The shape of the pairwise Pareto curves for each combination of objectives (see orange lines in Fig. 1) indicates clear trade-offs between production and carbon sequestration as well as between production and biodiversity. The nearly straight line that forms the frontier between production and biodiversity indicates direct trade-offs, meaning the increase in one benefit comes with a proportional decrease in the other benefit[51]. The slightly concave shape of the Pareto frontier between production and carbon sequestration implies that while there are clear trade-offs, there are scenarios in the centre of the curve where one benefit can be increased with a relatively small decrease of the other objective[51]. This means that increasing carbon sequestration from 0.8 to 0.9 will create a more substantial reduction in production than increasing carbon sequestration from 0.4 to 0.5, and vice versa. These differences show how crucial spatial targeting of land conversions is to minimise trade-offs. Examining the performance of the scenarios in terms of carbon sequestration and biodiversity in Fig. 1c shows clear synergies with both objectives increasing simultaneously. The frontier of Pareto-efficient scenarios for these two objectives is much shorter than for the other combinations. These strong synergies are caused by the substantial benefits that natural broadleaved forests offer for carbon sequestration and biodiversity. A comparison of the scenarios with a 100% weighting for carbon sequestration and a 100% weighting for biodiversity shows that they agree in 78.7% of the convertible cells on the same land conversions, which are mostly to natural broadleaved forest (in 78.3% of convertible cells) and to coniferous plantation forests (in 0.4% of the convertible cells).

Examining the land conversions that correspond to the points along the production-carbon sequestration Pareto frontier in Fig. 1a, frequent conversions from pasture to arable land and semi-natural habitats to pasture and some conversions from semi-natural habitats to plantation forests can be seen in the scenarios scoring high on production. However, the scenarios with higher values for carbon sequestration show higher conversion rates from arable land and pasture to semi-natural habitat. The 2D Pareto frontier for production and species occurrence (see Fig. 1b) shows similar conversions for the more production-focused scenarios. The scenarios with higher biodiversity performance show high conversion rates from pasture and arable land to semi-natural habitats, and some conversions from plantation forests to semi-natural habitats. For the very short frontier in Fig. 1c, the conversions in the Pareto-efficient scenarios are similar, with high conversion rates from arable land and pasture to semi-natural habitats. The main difference between the scenarios is that those scoring higher on biodiversity exhibit even higher conversion rates to semi-natural habitats from all land-use types, including conversions from plantation forests to semi-natural habitats, which are rarer in the carbon-focused scenarios. A table that shows the land conversion shares for all 231 scenarios and the spatial conversion patterns of four example scenarios (each prioritising one of the three objectives) can be found in Supplementary Table 2 and Supplementary Fig. 5.

### Current land-use and strictly better scenarios
Current land-use in GB achieves 0.62 of the potential land productivity in GB if production were the only objective. The current performances in biodiversity and carbon sequestration are 0.34 and 0.36 of what would be possible if land-use were targeted to maximise those objectives alone.

To identify the inefficiency of the current land-use in relation to the Pareto frontier, the distance between the normalised current performance and the frontier was measured parallel to each of the three axes. The point where the Pareto frontier is met when increasing production without changing the other two objectives has a production value 23.6% higher (+3.74 billion £ yr$^{-1}$) than the current (see Fig. 1a, b) without decreasing biodiversity or carbon sequestration. For the other two objectives, there is no intersection with the Pareto frontier when increasing one objective while keeping the other two constant. Instead, when increasing carbon sequestration while keeping production constant, biodiversity also increases. When increasing carbon sequestration as much as possible without decreasing production, the intersection point with the frontier shows an

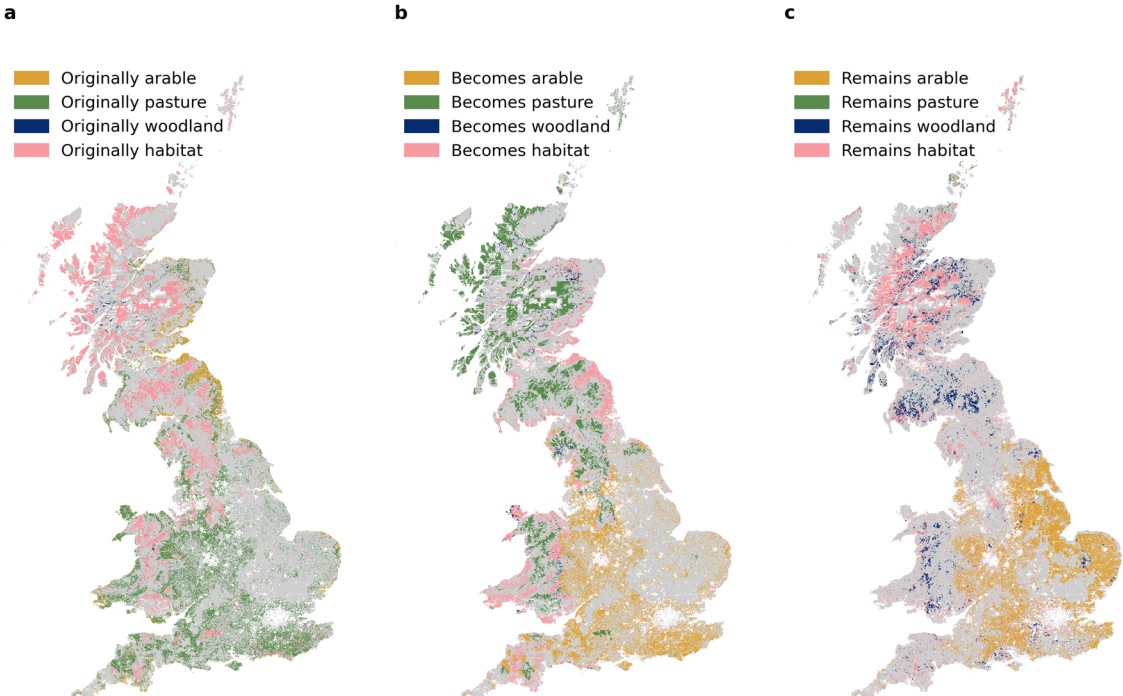

**a**

- Originally arable
- Originally pasture
- Originally woodland
- Originally habitat

**b**

- Becomes arable
- Becomes pasture
- Becomes woodland
- Becomes habitat

**c**

- Remains arable
- Remains pasture
- Remains woodland
- Remains habitat

**Fig. 2 | Common land-use conversions in all strictly better scenarios.** The eight scenarios that are strictly better than the current scenario all have the same changes on 33.24% of the convertible land **a** from the original land-use **b** to a new land-use in common, and **c** the same areas that remain unconverted in 31.19% of the convertible cells. Conversions are possible from arable land (yellow), pasture (green), plantation woodland (blue) and semi-natural habitat (pink).

improvement of +128.9% (an additional 71.9 million t $CO_2$-eq·yr$^{-1}$) compared to the current, shifting from substantial carbon emissions to a low level of carbon sequestration, and comes with an improvement of +22,530 biodiversity indicator points (see Fig. 1a, c). When increasing biodiversity as much as possible without decreasing production, the intersection point with the frontier shows an increase of 14.2% (+31,920 biodiversity indicator points) compared to the current biodiversity performance, which comes with an improvement of carbon sequestration of 37.9 million t $CO_2$-eq·yr$^{-1}$) (see Fig. 1b, c). The distances between the current performance and the points on the Pareto frontier are visualised in Fig. 1, and all normalised and actual numbers can be found in Supplementary Table 3.

To arrive at scenarios on the Pareto frontier from the current land-use, the conversion rates range from 45.0% to 72.0%, where the conversion rate is defined as the share of land that is converted. The most common conversion is from pasture to semi-natural habitat, which occurs in 17.6% of cells over all scenarios. In 6.9% of all scenarios, the most common change is from pasture to arable land; in 63.2%, the most common change is from pasture to semi-natural habitat; and in 29.9% of scenarios, we see a conversion of semi-natural grasslands to pasture.

Eight out of the 231 weighting combinations perform better or equal to the current state for all three objectives, meaning none of the three objectives would decrease. The conversion rate in those strictly better scenarios lies between 52.5 and 60.0%, which represents a substantial number of land conversions. The strictly better scenarios score between 0.45 and 0.63 of what would be possible on the land if carbon sequestration were the only objective (−34.65 million t $CO_2$-eq·yr$^{-1}$ to 9.27 million t $CO_2$-eq·yr$^{-1}$) compared to 0.36 (−55.76 million t $CO_2$-eq·yr$^{-1}$) under the current scenario. For production, the strictly better scenarios score between 0.65 and 0.76 of what would be possible (16.61 and 19.25 billion £·yr$^{-1}$), compared to 0.62 (15.8 billion £·yr$^{-1}$) under the current scenario. For biodiversity, the strictly better scenarios score between 0.34 and 0.45 of the maximum possible biodiversity score (225,000 and 245,000) compared to 0.34 (225,000) under the current scenario.

The eight strictly better scenarios contain the same land conversions in 33.2% of the convertible cells (see Fig. 2a, b). These include conversions to

arable land (35.5% of the common conversions) focused on south-east England (East and West Sussex, Kent, and Surrey), south-west England (Somerset, Wiltshire, and parts of Dorset), and the West Midlands. These areas are currently used for pasture and could potentially offer medium to high arable yields. Conversions to pasture account for 37.6% of the common conversions and occur mostly on semi-natural grassland (mostly acid grassland and heather grassland) and in areas with low potential arable yields and semi-natural habitat in areas that score comparatively low on biodiversity, such as central Wales and parts of Scotland. While conversions from semi-natural grassland to pasture are counterintuitive from an eco-logical perspective, they reflect the trade-offs under more production-focused priority weightings. One reason for this is the relatively low biodiversity benefit from semi-natural grasslands in our model. Additionally, most of these areas are likely already used for extensive grazing on acid grassland, heather, and heather grassland. Therefore, maintaining and expanding livestock grazing in these areas would come with relatively small disadvantages for biodiversity and carbon targets compared to the current pasture in southern England, which could offer valuable benefits as arable land or natural habitat. It is important to remember that the subset of strictly better scenarios does not allow a decrease in agricultural production and, therefore, also includes conversions to arable land and pasture where sensible. Considering the clear current prioritisation of agriculture over nature and the ambitious environmental objectives, the strictly better scenarios that do not allow any decrease in overall production might be considered unambitious, and slightly more biodiversity and carbon-focused changes might be considered appropriate.

Conversions to managed conifer forests make up only 2.32% of common changes and are mostly suggested on small areas of pastures in parts of Wales (Powys, Anglesey and Denbighshire) and small areas of semi-natural grassland in Cumbria and Perthshire in Scotland. The biodiversity and carbon sequestration benefits of natural broadleaved forests exceed those of coniferous plantations in most areas. Therefore, plantation forests are mostly seen in areas where a future semi-natural habitat type other than broadleaved forest is predicted. In 24.6% of the common conversions, land is converted to semi-natural habitat, mostly natural broadleaved forest. These

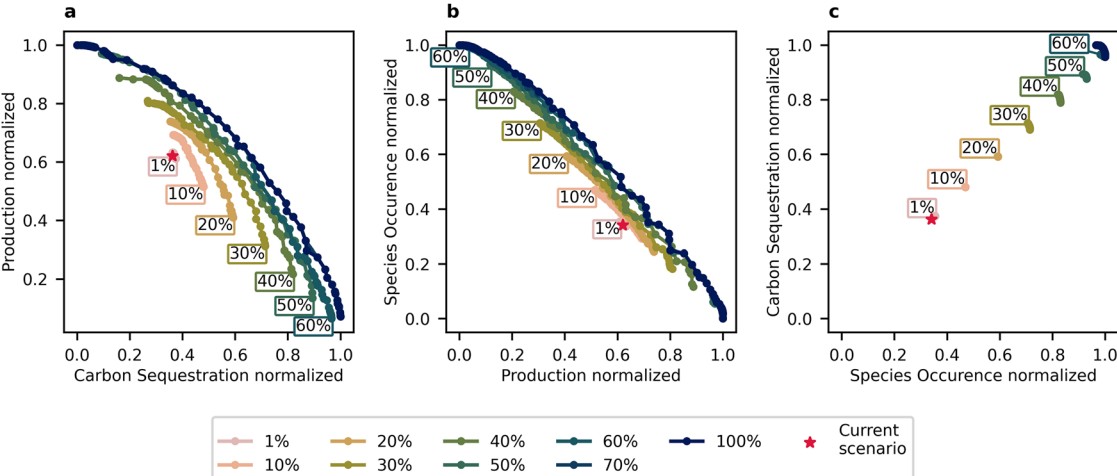

**Fig. 3 | Two-dimensional pairwise Pareto-efficient scenarios under a range of conversion budgets.** Two-dimensional pairwise Pareto contours for land conversion budgets from 1% up to 100% for trade-offs between normalised benefits for **a** carbon sequestration and production, **b** production and biodiversity, and **c** biodiversity and carbon sequestration. The colour scale indicates increasing conversion budgets, and the red star represents the current scenario.

conversions can be seen on small patches of pasture and less profitable arable land in England, especially in the north (Cumbria, Northumberland, Durham and Lancashire) and south-west (parts of Devon and south-east Cornwall). In Wales and in Scotland, especially in the south (South Dumfries and Galloway and the eastern part of the Scottish borders) and east (Angus, Fife and Lothian), most of the former pasture would be converted to natural broadleaved woodlands. Semi-natural habitats other than natural broadleaved forests are not seen in the common changes of the strictly better scenarios because they mainly benefit biodiversity without delivering any synergies for production or considerable carbon sequestration.

At the same time, all strictly better scenarios agree on keeping the existing land-use in the same 31.2% of the convertible cells (see Fig. 2c). These areas are mostly arable land, semi-natural habitats and existing conifer plantations; 44.3%, 39.7% and 14.3% of the areas that remain unchanged in all strictly better scenarios, respectively. Pasture that remains untouched in all strictly better scenarios barely exists (1.65% of cells). Arable land stays the same in most areas of England with comparatively high yields, especially in the East (Norfolk, Suffolk, parts of Cambridgeshire, Lincolnshire, Nottinghamshire and East Yorkshire), the West Midlands and the eastern parts of southwest England (Oxfordshire, West Berkshire, Wiltshire and Hampshire). Existing conifer forests remain largely untouched throughout GB. For semi-natural habitats, areas that remain unchanged include all existing land with fen and bog, all existing broadleaved forests in England, Wales, and Scotland, and heather and acid grassland in central Scotland.

## Conversion budgets—what can be achieved with a limited number of land conversions?

This analysis was carried out by limiting land-use change to a budget of change, meaning only a certain share of the land can be converted. For each combination of weights, we identify the ranked list of conversions across the entire country, based on the weighted sum of their performance. We then identify how many of those ranked conversions can be implemented within the total budget of conversions. Cells with the highest overall benefits under a weighting combination are chosen first, while less beneficial cells are added in merit order. The analysis was done for a range of conversion budgets. This enables decision makers to explore how much change would be necessary to achieve certain outcomes and, therefore, how ambitiously land conversion targets should be set. To explore a broad range of conversion rates, 11 budgets were analysed, starting with a budget of 1% and then stepping in increments of 10%, up to 100% of the rural land available for conversion. The resulting pairwise Pareto frontiers for the budgets are shown in Fig. 3. In Supplementary Fig. 6, the spatial conversion patterns of four example

scenarios (each prioritising one of the three objectives and one with a balanced weighting prioritisation) are shown for increasing budgets.

If, for a weighting, there is no further benefit from an increased budget because it would be best to maintain the existing land-use, then the actual conversion rate can be lower than the budget. Even with a budget of 100%, the highest conversion rate is 72.0%. Up to a budget of 40%, the full budget is used for all scenarios. From the 50% to 100% budgets, the lowest conversion rate is 45.0%.

The shape of the frontiers differs between the three objective pairings and across the set of budgets. In Fig. 3a, the frontiers, showing the relationship between carbon sequestration and production, have a slightly convex shape; only the 10% budget frontier is nearly linear, with the majority of each frontier pointing towards increased carbon sequestration. A reduction of carbon sequestration of more than 4% is only seen in budgets of 25% or above. The number of scenarios where both objectives improve (or do not decrease) is comparatively high even for the lower budgets, with 72 out of 110, 64/95, and 71/101 scenarios for conversion budgets of 1%, 5%, and 10%, respectively.

The pairwise Pareto frontiers for production and biodiversity (see Fig. 3b) have a similar linear shape and gradient for all budgets, so the trade-offs when only considering these two objectives stay nearly the same, independent of the chosen budget and location on the Pareto frontiers. This can be explained by the lack of synergies between the two objectives. Compared to Fig. 3a, c, the frontiers throughout the range of budgets are much closer together, meaning that a budget increase is not as beneficial for managing the trade-offs between production and biodiversity as for the other two combinations. This pattern is evident throughout all the budgets, with no clear turning points. Therefore, the number of scenarios where both objectives improve (or stay the same) is limited for all budgets, especially the lower budgets, with 6 out of 149 scenarios, 5/157, and 4/133 scenarios for conversion budgets of 1%, 5%, and 10%. The number of pairwise-strictly better scenarios is much smaller than for carbon sequestration and production. This indicates that the trade-offs between production and biodiversity are much larger, and land conversion choices that benefit both objectives are rare.

In Fig. 3c, the curves describing the relationship between carbon sequestration and biodiversity are very short, and therefore, show only minor trade-offs and almost no trade-offs up to a budget of 20%. For budgets of 60%, 70%, 80%, 90%, and 100%, the curves are closer to each other, and the additional benefit is, therefore, a lot smaller. For all budgets, all Pareto-efficient scenarios for carbon sequestration and biodiversity are also strictly better than the current situation for these two objectives.

The number of strictly better scenarios for all three objectives is very limited, with six out of 153 scenarios, 5/160 scenarios, 5/167 scenarios, and

**Fig. 4 | Frequency of change to each land-use seen per cell when overlaying all 231 scenarios.** The maps show the overall changes that occurred in each cell and the share of scenarios in which each cell is converted to arable land, pasture, plantation forest, and semi-natural habitat, respectively. The colour scale indicates the share of scenarios that a cell was converted in percent. The grey shading indicates that no conversion happened in any of the scenarios.

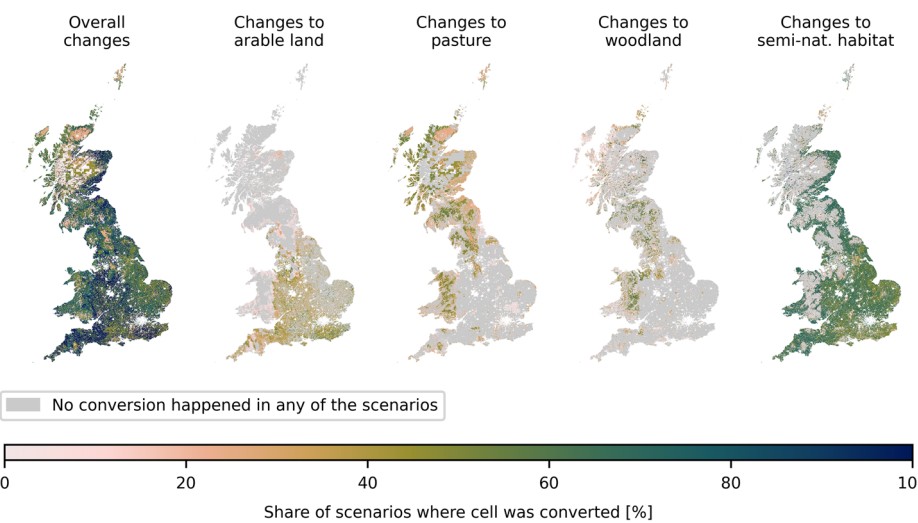

5/152 scenarios for the budgets 1%, 5%, 10% and 15% and eight strictly better scenarios for the 50%–100% budgets. This shows that, especially when changes are intended to be kept minimal, the careful selection of land conversions is crucial to avoid trade-offs. For all three objective pairings, the curves become increasingly long for higher budgets, meaning policymakers have a larger decision space to choose from.

### Identifying hotspots for change

To identify priority areas for land conversion, the scenarios with 231 priority weightings are combined by calculating the frequency of conversion in each cell across all scenarios. This allows for the generation of a heatmap of conversion frequencies. Cells converted under a broader range of weighting combinations can be considered hotspots for change, which are beneficial while showing comparatively smaller trade-offs. The more often a conversion occurs throughout the scenarios, the more priority weighting combinations it satisfies. It is therefore more robust to changing priorities. Figure 4 shows the relative frequency of change in each cell and the frequency of change to each land-use category, expressed as a percentage share of the 231 scenarios.

The most robust conversions to arable land can be seen on pasture areas in Southwest England (Somerset and North Somerset), Southeast England (East and West Sussex and Kent) and the West Midlands, with shares of 37–69% of the scenarios being converted in at least a quarter of the cells in each region. The most robust conversions to pasture occur in Wales (Gwynedd and Powys) and Scotland, where a quarter of the cells are converted in 33–59% of the scenarios. The most robust conversions to plantation forests occur in South Scotland, East Wales (Powys, Denbighshire and Wrexham) and Northwest England (especially Cumbria), where a quarter of the cells are converted in 4–100% of scenarios and the most robust 10% of cells are converted in 71–100% of scenarios.

Conversions to semi-natural habitats occur throughout England, with a share of 61–87% of scenarios being converted in the 25% most robust cells in each region, with lower values of 15–45% in areas with high arable yields or already existing plantation forests. In the southwest of Wales (Pembrokeshire, Carmarthenshire, and Ceredigion) shares of 68–75% of scenarios are converted in the most robust 25% of cells in each shire. Around the Scottish-English border and in the East of Scotland, even higher robustness values can be observed with shares of 68–100% of the scenarios being converted to natural habitat in the most robust 25% of cells. These are the areas in Scotland where broadleaved forests had been predicted as natural habitats.

### Priority grouping analysis

To summarise the broad range of scenarios and make them more intelligible, scenarios were classified into four categories. Scenarios that prioritise carbon sequestration (with a weighting ≥ 50 for carbon sequestration), scenarios that prioritise production (with a weighting ≥ 50 for production), scenarios that prioritise biodiversity (with a weighting ≥ 50 for biodiversity) and balanced scenarios (where all weightings are ≤50). All four groups include 66 scenarios. Scenarios where either objective is weighted 50 are included in both the balanced and the objective-focused group. The frequency of conversions in each cell over all scenarios in each group is shown in Fig. 5. In the group with carbon prioritising weightings, shown in the first row of Fig. 5, most changes are to semi-natural habitats in the form of natural unmanaged woodland and peatland restoration. The most common changes (in 90–98% of the scenarios) are conversions for restoration of lowland peat in eastern England (in Lincolnshire and Cambridgeshire), which is a particularly interesting area because it provides some high-quality agricultural land while, at the same time, in its drained form, is a substantial source of GHG. Equally common changes are conversions for the restoration of natural broadleaved woodlands in many parts of Wales, northern Devon, around the English-Scottish border (Dumfries and Galloway, Cumbria and Northumberland) and in East Scotland. Additionally, conversion to plantation forest occurs in 68–100% of all scenarios in the 10% most robust cells in each region in similar areas as broadleaved forest in Wales, South Scotland and Northwest England, where the predicted natural habitat is not broadleaved forest. Conversions to arable land are seen in a minority of scenarios (5–40%) all over England, with slightly higher values (14–53% of scenarios were converted in the 25% most robust cells in each shire) in East and West Sussex, and central and North Somerset. Conversions seen in all carbon prioritising scenarios (100%, dark blue shade) are limited to 2.17% of the convertible cells, with 2.04% of cells being changed to semi-natural habitat and 0.12% to managed plantation woodlands.

Very similar changes can be seen in the biodiversity prioritising group in the third row of Fig. 5. The areas prioritised for conversion to semi-natural habitats in the emissions-focused group also show high conversion rates (76–100% in the 25% most robust cells in each region) in the biodiversity-prioritising group. Only the conversion rate of arable land on drained peatland in Cambridgeshire does not increase as much compared to the emissions-focused scenarios, with values ranging from 85% to 95%. Conversions that all scenarios in the biodiversity group have in common are to semi-natural habitats in 20.7% of convertible cells. In Wales and Scotland, these include most areas where broadleaved forest is projected as rewilded habitat (in Pembrokeshire, Carmarthenshire, and Anglesey in Wales and along the eastern coast of Scotland and the English-Scottish border). In England, areas with lower expected arable yields would be converted in the north along the Scottish border and the east coast and in small patches in central England (parts of Gloucestershire, Northamptonshire, Bedfordshire, Hertfordshire and Northwest Essex), and the south-west.

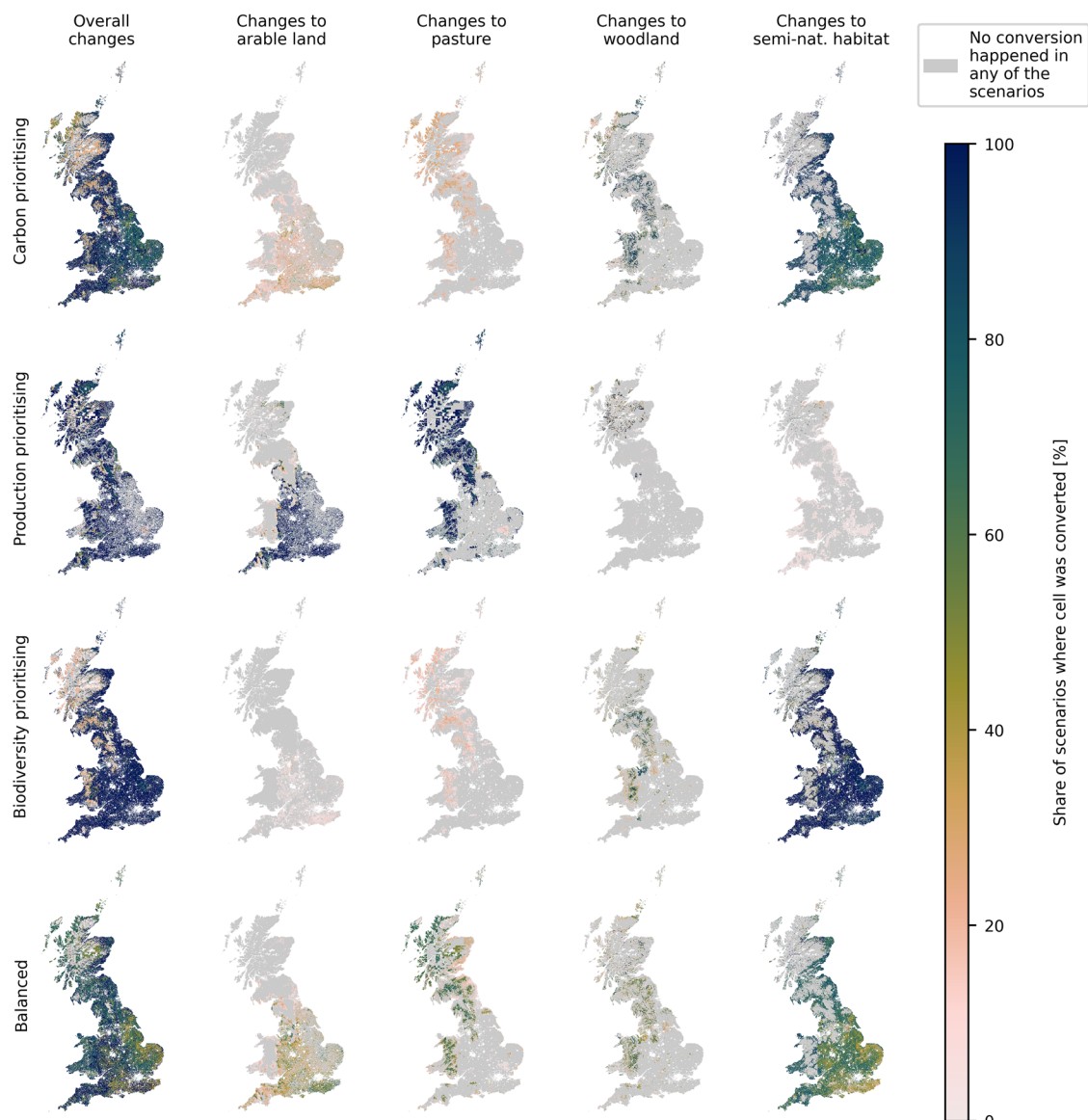

**Fig. 5 | Frequency of change to each land-use seen per cell in each of the priority weighting groups.** Frequency each cell was changed in general, to arable land, to pasture, to plantation forest, and to semi-natural habitat in the carbon prioritising, the production prioritising, the biodiversity prioritising and the balanced group. The colour scale proportion of scenarios where a cell was converted to each of the land-use types. The grey shading means that no conversion happened in any of the scenarios.

Changes observed in the production group primarily include conversions to arable land in most areas of England and expansions of pasture in Wales, Scotland, and northern England. Here, all scenarios agree on changes in 30.8% of the convertible cells. The same 12.7% of convertible cells are converted to arable land, especially in the south-west (Somerset, Dorset and Wiltshire), south-east (Sussex, Kent and Surrey) and in the West Midlands. The same 17.5% of convertible cells are converted to pasture, focusing on Powys and Gwynedd in Wales, parts of Northumberland, Cumbria, Durham and North Yorkshire in North England, and all over Scotland. A small share of 0.66% of convertible cells, mostly small patches in Cumbria and central and East Scotland, is converted to plantation forest in all production-focused scenarios.

Examining the balanced group, the share of convertible cells converted to the same land-use in all scenarios amounts to only 0.40% and includes small patches of plantation woodland and semi-natural habitat in Scotland. Changes to arable land in up to 83% of the scenarios occur in the south-east of England (Sussex, Kent and Surrey) and up to 86% in the southern part of Northwest England around Manchester. The most robust conversions to pasture occur in Scotland (near the border with England and along the East and North coasts), Central and North Wales, Northwest Yorkshire and North England (Cumbria, Northumberland and Lancashire), with shares of up to 70–77% of scenarios showing conversions. Conversions to coniferous plantation forests in more than 53–100% of scenarios in the 10% most robust cells per region are primarily seen in central Wales, North England and Scotland along the English-Scottish border. The most common land conversion in the balanced group is to semi-natural habitats, mostly natural broadleaved forest. These conversions occur in most of Wales (except central Wales, where acid grasslands are the projected natural habitat type), as well as in both South and East Scotland, and along both sides of the English-Scottish border. Additionally, many smaller patches of broadleaved forest with slightly lower conversion robustness are found throughout England, particularly in Southwest England (Devon and Cornwall), the East Midlands (Leicestershire and Northamptonshire), and the southern part of East England. The lower arable yields in these locations appear to be a more decisive aspect than differences in forest growth rates.

## Discussion

The multi-objective optimisation shown in this paper enables stakeholders to explore the full range of possibilities without any bias from predetermined (implicit) weightings. We thereby add to the literature of land-use scenario studies, which present a small number of optimised scenarios. Comparing all scenarios and determining the most common conversions enables us to identify beneficial land conversions that are robust to changing priorities. Optimising for a range of budgets helps to identify the most effective land conversions that should be prioritised if only a limited amount of land-use change is aimed for or if the success of policies encouraging change is uncertain. We find that the current land-use achieves 0.62 of the production possible on the land if this were the only objective, while scoring 0.34 for biodiversity and 0.36 for carbon sequestration if land-use were optimised to only those objectives, respectively. Using the distance between the current scenario and the Pareto frontier as a measure of the efficiency of current land-use, we find that the performance for each of the three objectives could be improved considerably. When maximising production without compromising biodiversity or carbon sequestration, we find a potential increase of 23.6%. When maximising carbon sequestration or biodiversity without decreasing any of the other benefits, increases of 128.9% and 14.2%, respectively, are possible. We find that the current land-use is far from the Pareto frontier and, therefore, far from optimal, independent of which objective weighting is considered.

A set of strictly better scenarios was identified, which all show conversions from pasture to arable land in England and to semi-natural habitats in Wales and Scotland while keeping existing arable land in most of England, as well as all existing plantation and semi-natural forests and peatlands. The conversion rate in those strictly better scenarios lies between 52.5% and 60.0% of all the rural land area in GB, which is relatively high and another indication of how far current land-use is from optimal. However, some scenarios, including the strictly better ones, show conversions from semi-natural grasslands to pasture, which might result from not including rough grazing on semi-natural grasslands in the analysis. If rough grazing were included, the conversion rates might be lower, and current production might score higher than 0.62.

The Pareto frontiers show strong synergies between biodiversity and carbon sequestration and near-linear trade-offs between production and biodiversity. The convex shape of the frontier between production and carbon sequestration shows that spatial targeting of land conversions is necessary to minimise trade-offs. The most robust land conversion is the planting of natural broadleaved woodland in current grassland locations that show relatively low potential arable yields. This is caused by its considerable benefits for carbon sequestration and biodiversity. The strong dominance of natural broadleaved forests over plantation forests, which are used for timber production, is mainly caused by the lower value for biodiversity and the life cycle of timber products. In this study, it is assumed that only 48.2% of carbon sequestered by timber plantations is stored long-term (based on recreating the calculations of Peng et al.[52] for the UK), which makes coniferous plantation forests less beneficial for carbon sequestration and storage than natural forests, notwithstanding conifer plantations' faster growth. Changes in the life cycle and usage of timber could change that and make plantations more valuable for long-term carbon storage. However, natural forests will remain the superior option because of the strong synergies they offer for biodiversity.

These findings align with other UK-focused land-use studies. One non-spatial study concludes that the pressure on land is high when trying to achieve several sustainability objectives while ensuring food security, but that policies that encourage more sustainable land-use could turn the land-use sector into a carbon sink[34]. Similarly, the CCC states that reducing land-based emissions is possible while achieving other objectives, such as food production and biodiversity, but will require radical changes in land-use[32]. While neither study is spatially explicit, they additionally include changes in consumption behaviour and management practices in their calculations, which have not been included in this study. The strong synergies we find for planting natural broadleaved forests in the right locations match the call for careful design of afforestation policies that should focus on creating natural forests instead of biodiversity-poor conifer monocultures[53] and in suitable locations[54].

While our model explores the interaction between ecological objectives and food production, it does not assume or enforce any specific dietary shifts. If current diets persist, especially in terms of their high consumption of animal products, which are very land-intensive, the conversion of agricultural land to other land uses comes with severe trade-offs. This is in line with other studies[55,56] which show that achieving climate and biodiversity targets without dietary shifts will be challenging. A reduction of land for food production, which occurs in many scenarios other than the most production-oriented, including the production of animal products, can, to some degree, be compensated by closing the efficiency gap. However, after this gap is closed, any further reduction will lead to an increase in imports, potentially from countries less committed to environmental targets or richer biodiversity, and, therefore, outsourcing of emissions and biodiversity degradation. Most land-use studies base their more sustainable scenarios on substantial dietary changes[34,57], which is a desirable objective and an essential step towards sustainability, but still uncertain. The National Food Strategy aims to tackle the "environmental damage caused by intensive agriculture"[58], but until then, an approach like the one presented in this paper can help focus land conversion on areas where trade-offs and losses in food production are minimised. Therefore, we highlight the importance of integrated policy approaches that combine land-use optimisation with a transformation of the food system, sustainable consumption, and minimisation of food waste to further decrease trade-offs and ensure sustainable outcomes. Additionally, it needs to be highlighted how the ratio between arable land and pasture, and therefore the share of animal products in overall production, differs across the scenarios. The current scenario has a ratio of 1.2 between land for arable production and pasture, while the Pareto-efficient scenarios show ratios between 0.11 and 3.0, with a mean value of 1.11.

Based on our results, several land-use interventions for GB can be identified. Overall, we find that biodiversity and carbon sequestration can both be improved substantially through land-use change without any decline in (food) production (see Fig. 1). However, if only a small amount of change is planned, it must be targeted at specific locations to avoid trade-offs. The conversion of current pasture in many parts of Wales and England plays a key role in achieving improved biodiversity and carbon sequestration without compromising food production. If increasing arable production would be of interest, land in the southeast of England that is currently used for pasture could offer land sparing high-yield production reserves (see Fig. 2). The best option for carbon sequestration and biodiversity is the planting of natural broadleaved forests, given the strong synergies between the objectives. If this is done in areas with relatively low potential arable yields, trade-offs are minimised.

The methodology chosen for this study aims to explore the trade-offs between carbon sequestration, production, and biodiversity from a biophysical perspective. However, local needs such as jobs or access to nature and cultural preferences must be considered in subsequent, more detailed studies aiming to design concrete policy implementation. Additionally, other factors might affect the feasibility of implementing the results of our analysis, such as land ownership or local political context and capacities. The economic costs of land conversion are another constraint on implementation that has not been included in this analysis. While the agricultural production calculated for this study provides an idea of where opportunity costs for farmers are higher in the case of conversions, it does not include the actual cost of new policy schemes for incentivising landowners, the cost of conversion to different land-use types, or other costs that might occur during implementation. For many land conversions, private costs exceed private benefits, and public funding will be necessary to enable these measures[21]. For instance, for planting broadleaved forests, the CCC estimated a net present value of −25,600 per hectare, including the costs for land purchasing, financing, planting, and establishment[21]. Higher conversion rates may be less feasible when these constraints are included.

Additionally, time delays from the time it takes for land uses and ecosystems to establish and deliver their full benefits have not been considered in this study. Future work could include these aspects to investigate which conversions are not only beneficial but also feasible when including factors such as land ownership and economic costs of land conversion.

As for most models, the results depend on the data input for the benefit calculations. Uncertainty in the input data includes a potential sampling bias in the biodiversity occurrence data, a lack of consideration for differences in soil composition beyond peaty soils in soil carbon storage, and potential spatial differences in forest age composition. Additionally, data for forest growth and agricultural yields are based on modelling and might include uncertainties about the realised local yields, especially under locally adapted management practices. Wherever possible, national datasets were used to enhance accuracy; however, no such dataset was available for arable yields. Thus, a global dataset had to be used instead. For the same reason, the spatially explicit woodland species composition data comes from a Europe-wide dataset. Therefore, our results should not be used to prescribe what exactly should happen in a specific location, but rather be interpreted as a general exploration of spatial trade-offs in GB. Additionally, benefit estimates can differ depending on the chosen indicators. Especially for biodiversity, using different indicators can lead to varying results. For computational reasons, it was impossible to include landscape connectivity in the biodiversity indicator, which might have led to a shift of the spatial patterns created, prioritising larger habitat patches and habitat corridors over isolated cells. Additionally, it is worth noting that to maintain independence between cells for the optimisation process, biodiversity was represented on a cell-wise basis. As such, the measure does not consider the relative degradation of different broad ecological systems or non-linearities arising from large-scale condition changes that interact with this at the landscape scale. This limitation is not unique to this study.

Grouping all semi-natural habitats into one category without allowing conversions between them leads to lower conversion rates to semi-natural habitats in areas with habitat types with comparatively lower benefits. The habitat sub-category within the semi-natural habitat class to which a land cell would be converted was chosen based on a nearest-neighbour algorithm combined with a prioritisation of peatlands on peaty soils. Allowing the conversion of these habitats to better-performing habitat types, especially natural broadleaved forests, would have led to slightly higher conversion rates to semi-natural habitats and lower conversion rates to plantation forests. At the same time, converting most existing semi-natural habitats to broadleaved forests is not the right approach for biodiversity either, since a broad range of different semi-natural habitats is needed.

We have assumed urban land-use to remain unchanged, and it has not been evaluated for any of its benefits. However, urban expansion and housing development are predicted to occupy additional land in the UK[59], and some argue that this should occur, especially on the Green Belts surrounding existing urban areas[60]. That will require the conversion of current rural areas, which puts additional pressure on the land and affects the feasibility of our model outputs in these areas. To minimise this uncertainty and an additional source of trade-offs, it has been recommended to keep new urban development compact[34]. These effects are currently not part of the model but can be included with urban expansion scenarios in a future version. Furthermore, we did not exclude protected areas and other classified areas from consideration for land conversions to explore the full theoretical space of options without imposing current institutional constraints. Protected areas, for example, can include arable land or pasture and are relevant for our trade-off analysis. However, we acknowledge that this might lead to an overestimation of the feasibility of land conversions in certain areas. Future work could include these institutional and legal limitations.

Additionally, land suitability may change under future climate change, especially for agricultural production[61]. These potential future changes have not been included in this static analysis, but subsequent research will incorporate future climate scenarios into the modelling framework to enable more long-term planning. Another aspect that could be included in future

work is the differentiation of arable land (extensive or intensive) and land management practices, which might mitigate trade-offs, such as organic farming or agroforestry. This also applies to other adjustments of land management and agricultural practices, including how livestock is managed and whether it is primarily grass-fed or if arable land is used for animal feed production. This is also in line with discussions about land sharing versus land sparing, which could be explored as well, even though a study conducted in Asia suggests that land allocation might have a more meaningful influence on achieving a set of ecosystem services and food production than sharing versus sparing[62]. Land ownership should be considered to assess the feasibility of land-use change in a cell, which can be explored in future work.

## Conclusions

To address the issue of the triple challenge of climate mitigation, biodiversity, and food production, a multi-objective analysis has been implemented to explore land-use trade-offs and synergies between (food and forestry) production, carbon sequestration, and biodiversity, and identify the most effective and robust place-specific land conversions. Investigating a wide range of priority weightings allows stakeholders to explore the full range of possible land-use changes in GB, avoiding the potential biases associated with evaluating only a handful of scenarios or applying an implicit weighting by translating the benefits into one common unit (such as the monetary value). Complementing the presentation of trade-offs in the form of Pareto frontiers with a representation of changes seen throughout all the scenarios identifies the most critical locations for land-use conversion. Running the model with different total areas of permitted conversion (conversion budgets) allowed identification of the locations that should be targeted if only a small amount of rural land-use conversion is politically feasible. The same concept and methodology could be beneficial for exploring land-use trade-offs from all kinds of land conversions, including other land-use types such as energy crops or the development of new housing and suburban areas. With sufficient data, it can be done in any country or scale, from regional to global.

Our results show strong synergies between carbon sequestration and biodiversity and clear trade-offs between food production and biodiversity. Nevertheless, it is possible to considerably increase biodiversity and carbon sequestration from today's land-use configuration without adverse effects on food production. However, the number of scenarios where all three objectives increase or stay the same is limited, so it will require very targeted policies and incentives. The best choice for improving biodiversity and carbon sequestration while exploiting synergies between both is by creating natural broadleaved woodlands in the right locations. If this is done in places with low potential arable yields, trade-offs with production can be minimised.

## Methods

A land allocation and optimisation approach for different budgets of land conversion was used to explore the decision space for change and identify trade-offs between the three objectives carbon sequestration, biodiversity, and production in GB. A resolution of 500 × 500 m was chosen to allow the representation of landscape characteristics while keeping the computational and data requirements reasonable. This results in our model containing a total of 814,004 convertible grid cells. Each grid cell can be converted into any of four land-use types: arable land, pasture, plantation forest, and semi-natural habitat. These types represent the following distribution of primary land-use types of convertible land: 25.77%, 31.59%, 6.49% and 36.15%, respectively. We assume that conversions between different semi-natural habitats are less relevant when evaluating land-use trade-offs; therefore, they were grouped under one category. The categories are comprised of a combination of sub-classes based on the land cover map categories used by the UK Centre of Ecology & Hydrology (UKCEH) (see Table 2).

The associated potential benefits of land conversions were pre-calculated based on the characteristics of each cell's location, resulting in benefit potential maps. Based on these potential benefits, the optimal conversions and scenarios for a range of priority weightings were calculated

using a weighted sum multi-objective optimisation (see Fig. 6). The resulting Pareto-frontier of non-dominated scenario performances and the corresponding spatial scenarios were then used to analyse the trade-offs between the three objectives and identify the location-specific conversions that offer the most synergies.

## Calculation of benefits

Static maps were created showing the potential benefits of converting each land cell to each of four land-use classes, arable land, pasture, plantation forest and semi-natural habitat, and for each of the three performance indicators, respectively. For each combination of objective weightings, the optimal land conversions were selected for each cell based on the potential benefit in the location. For each created scenario, the optimised benefit values of all cells were added up to calculate the overall performance of the land configuration.

The Land Cover Map 2020 (25 m rasterised land parcels, GB) published by UKCEH[63] was aggregated to a resolution of 500 m using a nearest-neighbour resampling approach, which allowed for maintaining the original shares of land-use classes (see Supplementary Fig. 7). This was used as a baseline scenario for potential land conversions. Cells with the following land cover types in the UKCEH land cover map[63] are not available for potential land conversion and were therefore excluded: inland rock, salt- and freshwater, supralittoral rock and sediment, littoral rock and sediment, saltmarsh, as well as urban and suburban areas. The remaining cells, which are considered available for potential land conversion, were categorised as either arable land, pasture, plantation forest, or semi-natural habitats based on the current land cover. An exclusion of land for conversion based on land classifications such as national parks, battlefields, etc., was not considered to explore the full theoretical space of possibilities without any current institutional constraints. This leaves 86.82 % of the land area available for potential change. For each cell, the benefits in terms of emissions, production, and biodiversity benefits were calculated for each potential conversion using the data shown in Table 3 and captured in 12 benefit maps (4 land-use categories for conversion x 3 benefits). To ensure consistency, all input datasets were resampled to the 500 × 500 m resolution of the optimisation grid using a nearest-neighbour or bilinear resampling algorithm. We also conducted a sensitivity analysis to evaluate the impact of different resampling strategies on the input data. While resolution mismatches were minimised, we acknowledge that some residual effects may remain and could influence outcomes in areas with high spatial variability. To keep the computational effort feasible, benefits from each land cover type for each cell were assumed to be fixed instead of using a dynamic model, and where appropriate (e.g., for forestry), benefits were aggregated over a lifecycle. Therefore, a benefit for GHG sequestration, production, and biodiversity was calculated for each cell and each possible change of land-use in that cell.

For calculating the carbon sequestration benefit of potential land conversions, carbon storage values for soil and vegetation were used from Cantarello et al.[64]. Each of the included land cover types from the UKCEH land cover map was assigned the corresponding value from Cantarello et al.[64]. The carbon emissions or sequestration from each possible conversion was computed by calculating the difference between the current carbon storage and the potential future storage. For the semi-natural habitats, a more differentiated approach was chosen to consider the differences in carbon sequestration and storage between different habitat types. To project the most likely habitat type for each cell that would be selected when restoring nature, a nearest neighbour analysis was used (explained in more detail below). For

**Table 2 | Land cover categories used in the model and the corresponding UKCEH land cover subcategories**

| Land-use category | UKCEH Land cover types |
|---|---|
| Arable | Arable and horticulture |
| Pasture | Improved grassland |
| Plantation forest | Coniferous woodland |
| Semi-natural habitat | Neutral grassland<br>Calcareous grassland<br>Acid grassland<br>Heather grassland<br>Fen marsh and swamp<br>Heather<br>Bog<br>Broadleaved woodland |

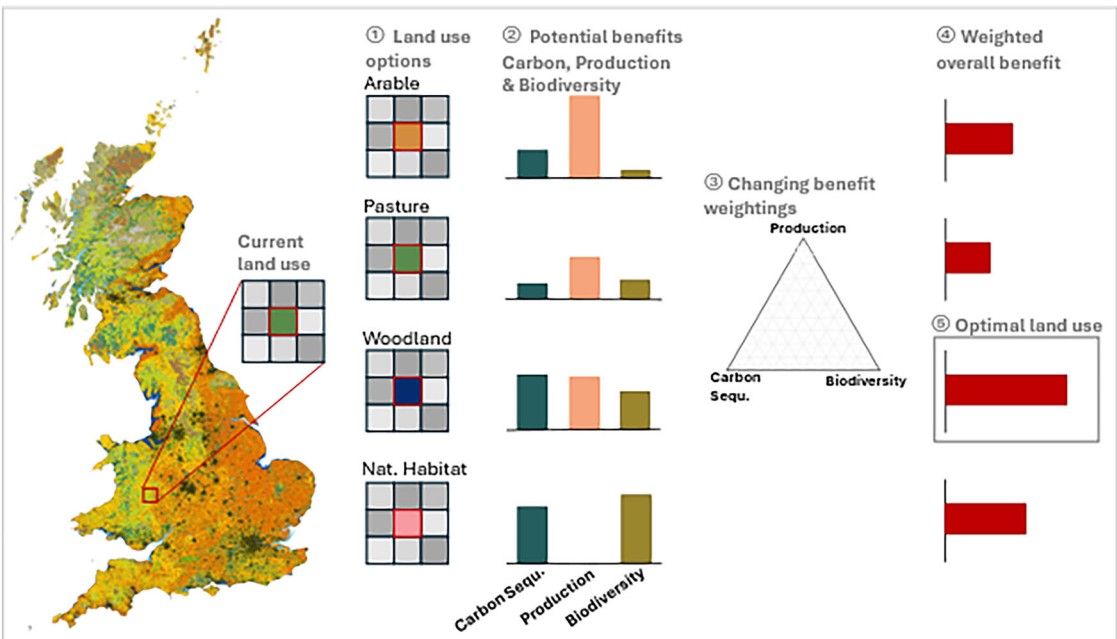

**Fig. 6 | Overview of the steps in the methodology.** ① Conversions from the current land-use are possible to four land-use categories. ② The potential benefits of conversions to those four land-use categories are calculated against three objectives: carbon sequestration, agricultural/forestry production and biodiversity. ③ The potential benefits are weighted using 231 weighting combinations ④ and summed up to the resulting weighted overall benefit. ⑤ The highest weighted overall benefit and the corresponding optimal land-use conversion are identified.

**Table 3 | Data used to create the benefit maps for the three objectives, carbon sequestration, production, and biodiversity, as well as the four land cover categories: arable land, pasture, plantation forest, and semi-natural habitats**

| | Carbon sequestration | Production | Biodiversity |
|---|---|---|---|
| Arable land | - Carbon storage of and flows between land cover classes[64]<br>- Emissions from fertilisers[65] | - Crop distribution (physical and harvested areas)[85]<br>- Agricultural yields[86]<br>- FAO stats producer prices[87]<br>- FAO stats production areas[88] | - UKCEH land cover map[63]<br>- Species occurrence data[99]<br>- Climatic data[101]<br>- JNCC Species distribution model ensemble<br>- Habitat condition values based on PREDICTS biodiversity data[106] |
| Pasture | - Carbon storage of and flows between land cover classes[64]<br>- Emissions from livestock[69] | - FAO stats livestock patterns[66]<br>- Shares of beef and dairy cows[68]<br>- FAO stats producer prices[87]<br>- Livestock yields[88]<br>- FAO pasture suitability indicator[89] | |
| Plantation forests | - Carbon storage of land cover classes[64]<br>- Woodland species composition[76,77]<br>- Location-specific yield classes[78]<br>- Woodland carbon sequestration[79]<br>- Tree age distribution[81]<br>- Wood products data[82–84] | - Woodland species composition[76,77]<br>- Location-specific yield classes[78]<br>- Woodland carbon sequestration[79]<br>- Tree age distribution[81]<br>- Tree carbon densities[91]<br>- Timber prices[92] | |
| Semi-natural habitats | - Carbon storage of land cover classes[64]<br>- Carbon fluxes for different states of peatland[75]<br>- peaty soils for England[71,72], Scotland[73] and Wales[74]<br>- (for broadl. woodlands, the same data sources as for plantation forests were used, except[82–84]) | - No production is assumed | |

arable land emissions from fertilisers, an average of 1460.67 g $N_2O$-N·ha$^{-1}$ was assumed based on the emissions reported by Bell et al. [65], which converts to 626.63 kg $CO_2$ eq·ha$^{-1}$. Differences between management practices or soil composition have not been included.

To calculate the emissions from pasture, the average livestock density per meadow and pasture area between 2016 and 2020 was taken from the United Nations Food and Agriculture Organisation's (FAO) database FAOSTAT[66] and converted using the livestock unit coefficients used by Eurostat[67]. Shares for beef and dairy cows from the UK gov livestock statistics[68] were applied. Emission values for beef cows, dairy cows, and sheep for $CH_4$ and $N_2O$[69] were averaged and converted to $CO_2$-eq using conversion values for a global warming potential over 100 years (GWP-100)[70]. Emissions from livestock per ha on pasture sum up to 7.62 t $CO_2$ eq·ha$^{-1}$·yr$^{-1}$, assuming an average livestock mix on all pasture areas. All detailed numbers are provided in Supplementary Table 4. In reality, beef cows, dairy cows and sheep would not be grazing on the same land patch and stocking densities and management practices might differ substantially between farms and regions.

To include the substantial influence of drained and intact peatlands, areas with peaty soils were identified using data on existing peatland and peaty soils for England[71,72], Scotland[73] and Wales[74]. Then, emissions values were assigned according to the peatland's current and potential future state[75]. Drained peatlands used for farming are a considerable source of GHG emissions, which decreases when bogs and fens are rewetted. Undrained, near-natural fen presents an important carbon store that also sequesters small amounts of carbon.

To calculate the sequestration and emissions from forest plantations and natural broadleaved forests, tree species occurrence maps from the European Forest Institute[76,77] were obtained for six coniferous and seven broadleaved tree species. These were combined with location-specific yield maps from the Forest Research Ecological Site Classification[78] and the cumulative total sequestered carbon values per yield class from the woodland carbon code[79]. No thinning is assumed for broadleaved forests, while thinning is assumed in the management of coniferous plantations. Biomass removed due to thinning is added to the cumulative sequestration. For managed plantation forests, felling times were assumed based on the tree and yield class-specific age of maximum mean annual volume increment[80]. The time scale of carbon sequestration and the resulting differences between already established forests and new conversions to forests are included by considering the tree age distribution[81], the species and yield specific growth rates and for plantation forests the resulting time until felling. The age

distribution of newly planted forests is assumed to result from consistent planting between the baseline year 2020 and 2050. All assumptions for newly planted and existing broadleaved forests, as well as coniferous plantations, are summarised in Supplementary Table 5. To estimate the ratio of long-term carbon storage in harvested wood products, the global wood product model by Peng et al. [52] was recreated for the UK using the FAO-STAT Forestry Production and Trade data for the UK in 2022[82] and the Forest Research Forestry Statistics 2023[83,84]. Average yearly numbers for carbon sequestration/emissions from plantation woodlands and natural broadleaved forests were calculated over the period 2020–2050. For plantation woodlands, this number is then multiplied by the share of long-term carbon storage in harvested wood products. Our assumptions are subject to a number of uncertainties regarding differences in management practices, harvesting patterns, the exact tree species composition and the choice of tree species being planted in afforestation projects.

Production includes food production from arable land and pasture, as well as timber production from forest plantations and is calculated as average yearly revenue from 2020 to 2050. For priority habitats, no production is assumed. For estimating food production on arable land, harvested and physical areas of production for eight different crop groups (wheat, barley, rapeseed, sugar beet, potatoes, vegetables, other cereals and other pulses) were obtained from the MapSPAM dataset[85]. Since the overall reported areas did not always fully match the arable land area in this model, the area shares of each crop group in each location were calculated. Attainable crop yields for eleven crops (wheat, barley, rapeseed, oats, beans, potatoes, sugar beet, peas, carrots, cabbage and onion) under rainfed conditions from FAO GAEZv4[86] were converted from dry weight to wet weight using the conversion factors from the model documentation[86] and matched with the MapSPAM crop groups. Food prices for each crop group were computed by calculating the mean FAOSTAT producer prices in £·ha$^{-1}$ for the period of 2017 − 2021[87] and weighting them by the percentage share of that crop of the total harvested area[88]. Extreme outliers > 1.5 IQR were removed and replaced with the mean value of the surrounding cells

For food production from pasture, the average production of milk and meat from cattle, as well as meat and skin from sheep per hectare, was calculated based on calculated livestock densities[66], shares of beef and dairy cows[68] and yields[88] and multiplied by the average producer prices over 2017 − 2021[87]. To identify areas that are unsuitable for pasture, the FAO pasture suitability[89] was used and all areas with a suitability index under 20 were assumed to bring no returns of production[90].

For timber production, the forest model described above was used. Since the Woodland carbon code data does not include wood biomass production, the carbon sequestration data were converted into tree mass using carbon densities for the considered tree species[91]. Timber harvesting was assumed to occur at the age of maximum mean annual volume increment[80] and the timber price was calculated using the average price over the years 2018–2023[92]. Considerably lower returns from freshly planted woodlands were considered by including the location- and species-specific rate of forest growth and, based on it, the time until the first harvest.

To project the habitat type that each cell would be converted to were it to be restored to semi-natural habitat, a nearest neighbour analysis was performed. To ensure that peatland areas were included on all peaty soils but only on these, peatlands were excluded in the nearest neighbour analysis. Then, existing peatland and peaty soils for England[71,72], Scotland[73] and Wales[74], bog and fen from the UKCEH land cover map[63] were added to the layer. Therefore, peatlands were prioritised over the results of the nearest neighbour analysis since peatland restoration is crucial for biodiversity and climate mitigation. Then, cells classified as one of the semi-natural habitats in the UKCEH land cover map were added to the map

Here, we used a combination of species occurrence probabilities from a species distribution model (SDM)[93] and a land-use specific area-equivalent habitat condition score derived from the PREDICTS database[94,95]. The species distribution model was developed in line with Croft et al.[96] using 86 species from the UK Biodiversity Action Plan priority species list[97], including 38 vascular plants, 13 mammals, 22 invertebrates, five lichens, one herptile and seven birds. A detailed list of all species can be found in Supplementary Table 6 in the SI. This selection was based on a list of priority species that have also been used in the SDM run as part of the NEVO land-use model[42] and was composed based on a range of criteria, including the number of records in the last 20 years, the spreading of a species and independence from water quality[98]. The species occurrence data was downloaded from the NBN atlas database[99] (for a list of the NBN atlas data partners, see Supplementary Table 7), which has some limitations due to it not being systematically surveyed, but is the most comprehensive dataset available. As environmental variables, 19 bio-climatic variables were calculated in R using the biovars function in the R package dismo. Therefore, monthly rainfall, maximum temperature and minimum temperature data from 1999 to 2021 from the HadUK Climate Observation data were downloaded from the Centre for Environmental Data Analysis Archive[100,101] with a resolution of $1 \times 1$ km. Land-use-based environmental variables were included as a share of each land cover type within a $1 \times 1$ km cell based on the UKCEH land cover map[63]. The SDM was run using the Boosted Regression Tree and the Random Forest models from the JNCC SDM suite ensemble (available on https://github.com/jncc/sdms). The best-performing model for each species was chosen based on the area under the curve measure. After training the model, the response to land cover change was simulated by keeping the climate variables constant but changing the land cover shares in each cell. The resulting occurrence probabilities for all species were summed up to an overall indicator ($\sum_{species} P(\text{cell, species})$, where $P$ is the occurrence probability), as suggested by Calabrese et al.[102]. The resulting indicators were then normalised over all land-use categories to values between 0 and 1.

To incorporate the inherent value of natural habitats that might naturally be less species-rich, land-use specific habitat condition scores consistent with those used in the global Biodiversity Habitat Index[103,104] were attributed to each land-use class. For each land-use, the proportion of native species[105] was extracted from the PREDICTS database[95,106] and rescaled using the species-area relationship[107,108]. The resultant condition metric is scaled from fully intact (1) to fully degraded (0) in units of effective area of habitat, which can be summed over a region. Arable land was classified with a condition value of 0.08, pasture with 0.1, plantation forest with 0.23, grasslands with 0.3, and all other semi-natural habitats, including broad-leaved forests (secondary) with values between 0.33 and 0.7, depending on the type of habitat and if it already exists or would be newly established. The higher habitat condition values given to established secondary semi-natural habitats (0.38–0.7, depending on the habitat type) compared to newly created ones (0.33) ensure that the increased ecological value of a well-established natural area and the lower ecological value of a freshly converted habitat are reflected in the model. The species occurrence projection was normalised to values between 0 and 1, and then both indicators were combined by calculating their geometric mean.

## Optimisation and analysis

The resulting benefit maps were used as the basis for a multi-objective optimisation. Optimisation techniques are a common approach for land allocation and decision-making problems[109–111]. Instead of expressing all objectives in one unit, such as monetising them or finding one ideal solution by weighting the objectives, multi-objective optimisation produces a Pareto frontier of non-dominated scenarios. Pareto-efficient or non-dominated means it is impossible to improve one of the objectives further without decreasing one of the other objectives, consequently.

As the aggregate performance with respect to the three objectives is a linear combination of the performances at every grid cell, an exhaustive search on a regular grid can be used to calculate the full range of possible objective weightings to cover the full Pareto frontier. Therefore, 231 weight combinations with a step size of 5 were used, where 100%-0%-0% would fully prioritise the first objective while 35%-35%-30% would consider all three objectives nearly equally. The step size of five was chosen as a trade-off between resolution and the comprehensibility of the number of scenarios, as well as the computational runtime. A smaller step size would have increased the number of scenarios considerably, making exploration infeasible and increasing the computational runtime. Conversely, a larger step size would have led to excessive discretisation, making it impossible to draw the full Pareto frontier (see Supplementary Figs. 1–4). Preliminary tests showed that a step size of five was able to create a smooth Pareto frontier while keeping the number of scenarios manageable.

The steps of the optimisation are visualised in Fig. 6. After calculating the location-specific potential benefits of conversion to each of the four land-use categories (as described above), the benefit values are normalised for each benefit across all four land-use categories.

Subsequently, for each constellation of weights, the following steps were implemented to create a scenario maximising the overall weighted benefit:

With $w_O$ as the weight of each objective $O$ in the scenario, $O = C, P, B$ (carbon sequestration, production, biodiversity) where $\sum_O w_O = 1$, it was calculated how high each of the land-use types $k$ scores in each cell $n$ in terms of the overall weighted benefit $b$:

$$b_{n,k} = \bar{b}_{n,k}^C \times w_C + \bar{b}_{n,k}^P \times w_P + \bar{b}_{n,k}^B \times w_B \qquad (1)$$

Where $\bar{b}_{n,k}^O$ is the normalised benefit to objective $O$. Then, for each cell $n$ the LU type $k$ that yields the maximum weighted benefit is chosen:

$$k_n = argmax_k b_{n,k_n} \qquad (2)$$

$k_1, k_2, k_3, \ldots, k_M$ together define a scenario which maximises the overall weighted benefit $\sum_n b_{n,k_n}$.

For each of the created scenarios, the benefits for carbon sequestration, production and biodiversity were calculated, and the Pareto-efficient scenarios were conserved. The trade-off between the three objectives (normalising benefits to 0–1) was assessed by identifying the overall shape of the three-dimensional frontier and the gradient and curvature of the curves describing the Pareto frontiers between each combination of two objectives. The shape of the resulting Pareto frontier can be analysed to understand the relationship between the objectives. While a straight line represents a direct trade-off where the increase of one benefit leads to a proportional decrease of another benefit, a concave curve implies that while there are trade-offs, it is possible to increase one benefit without seeing a proportional decrease in the other service[51]. Additionally, the non-dominated scenarios were compared

**Fig. 7 | Summarising scenarios in four categories based on their objective weightings.** Scenarios are grouped based on their priority weighting under Carbon prioritising (carbon sequestration weighted ≥ 50), Production prioritising (production weighted ≥ 50), Biodiversity prioritising (biodiversity weighted ≥ 50), and Balanced (all three objectives weighted ≤ 50).

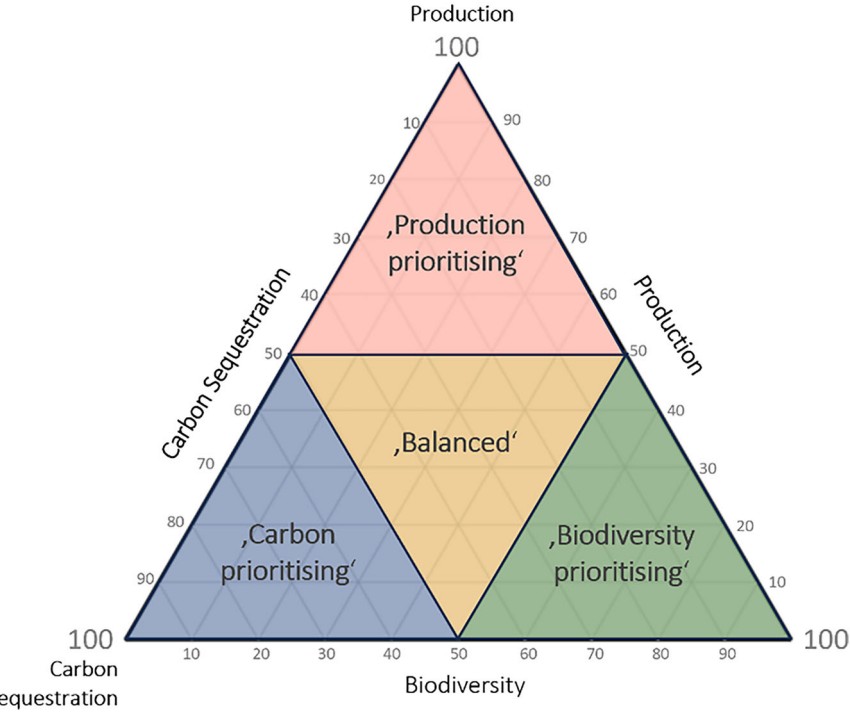

to assess the importance of specific locations in terms of trade-offs resulting from land conversion.

Realistically, not all rural land considered in this model could be converted to a new land-use type. Converting land to new land-use comes with many challenges, and planning with high rates of change might be politically infeasible. Therefore, an approach is needed that allows for the exploration of the potential benefits of limited land conversion and the marginal benefits of increased percentages for the area converted. To identify the areas that should be prioritised when converting land and designing land-use policies, scenarios were calculated for a range of conversion budgets, starting from 1% to 100% of the available area. This approach chooses the land conversions first that offer the highest cellwise benefits per area and are, therefore, the most efficient. Additionally, it allows us to explore the decision space for change, demonstrate how much change will be necessary to attain certain objectives, and identify a threshold where additional change adds only minimal additional benefits. Therefore, the same weighted approach as described above was used, but cells are converted successively, from most to least beneficial, until the change budget is met. Where more cells than were comprised in the budget were equally beneficial, cells for conversion were selected randomly.

To spatially identify trade-offs and synergies, all 231 scenarios were compared, and the percentage of scenarios where each cell was converted to a specific land-use cover was calculated and compared between budgets and land-use types. If certain changes came up in a broader range of scenarios, and therefore priority weightings, these conversions were considered more robust to changing priorities of decision-makers and had fewer trade-offs. If, in contrast, the suggested conversion for an area differed a lot depending on the chosen priority weightings, the conversions were not considered robust and are likely to show higher trade-offs. While comparing the maps for different priority weightings can help identify conversions that are robust to changing priority weightings, looking at the budgets facilitates identifying the most effective land conversions

To help summarise the findings and make use of the full range of scenarios, they were grouped according to their weighting as either carbon prioritising (carbon sequestration weighting ≥ 50), production prioritising (production weighting ≥ 50), biodiversity prioritising (biodiversity weighting ≥ 50) or balanced (none of the three objectives is weighted > 50) as

shown in Fig. 7. The scenarios along the boundaries, with a weighting of 50 for one of the objectives, were included in the balanced and objective-focused group. Then, the four priority groups were compared, and the changes most common in the range of scenarios were analysed.

### Reporting summary
Further information on research design is available in the Nature Portfolio Reporting Summary linked to this article.

### Data availability
All datasets used in this study are cited in the relevant sections. The land cover raster can be downloaded from the UK Centre for Ecology & Hydrology via the EDINA Environment Digimap service (https://digimap. edina.ac.uk/environment) or https://www.ceh.ac.uk/data/ukceh-land-cover-maps. Spatial data for drained peatlands and peaty soils were downloaded for England from Natural England via https://www.data.gov. uk/dataset/9d494f48-f0d7-4333-96f0-8b736ac8fb18/peaty-soils-location1 and https://www.data.gov.uk/dataset/b12f420a-d9f1-4966-aa3e-0f6e680e3875/moorland-deep-peat-ap-status1, for Wales from UKCEH via https://catalogue.ceh.ac.uk/documents/58139ce6-63f9-4444-9f77-fc7b5dcc00d8 and for Scotland from Scottish Natural Heritage via https://opendata.nature.scot/datasets/snh::carbon-and-peatland-2016-map/explore. Tree species maps for European forests were downloaded from the European Forest Institute via https://efi.int/knowledge/maps/treespecies. Tree species and location-specific yield class potential data can be obtained from the Forest Research Ecological site classification tool via http://www.forestdss.org.uk/geoforestdss/esc4.jsp. Yield class specific carbon sequestration data was obtained from the Woodland Carbon Code Lookup tables in the Carbon calculation spreadsheet, which can be downloaded via https://www.woodlandcarboncode.org.uk/landowners-apply/template-documents. Forestry production and trade data were downloaded from FAOSTAT and found here: https://www.fao.org/faostat/en/#data/FO. The MapSPAM raster data on crop production areas was downloaded via https://mapspam.info/. Location-specific attainable arable yields can be downloaded from FAO GAEZv4 via https://gaez.fao.org/pages/data-viewer. Statistics on livestock patterns can be downloaded from FAOSTAT via https://www.fao.org/faostat/en/#data/EK. Producer prices for agricultural

products can be downloaded from FAOSTAT via https://www.fao.org/faostat/en/#data/PP. Total harvested area and average livestock yields can be downloaded from the Crops and livestock products dataset from FAOSTAT via https://www.fao.org/faostat/en/#data/QCL. The Suitability of global land area for pasture (FGGD) raster data can be downloaded from FAO via https://data.apps.fao.org/catalogue/iso/2b357400-891a-11db-b9b2-000d939bc5d8. Data on carbon densities for different tree species can be downloaded from the Global Wood Densities Database via https://datadryad.org/stash/dataset/doi:10.5061/dryad.234. The species occurrence data were downloaded from the NBN atlas database via https://nbnatlas.org/, the Data Provider, Original Recorder, and the NBN Trust bear no responsibility for any further analysis or interpretation of that material, data and/or information. The NBN atlas data partners that collected the species occurence data are listed in Supplementary Table 7 which can be accessed in its full length via https://figshare.com/articles/dataset/CarbonFoodNature_TradeOffs_-_Source_data/29618120. Rainfall and temperature data from the HadUK-Grid Gridded Climate Observations data can be downloaded from the Centre for Environmental Data Analysis (CEDA) Archive via https://catalogue.ceda.ac.uk/uuid/bbca3267dc7d4219af484976734c9527/. Data on the proportion of native species for the habitat condition calculations can be downloaded from the PREDICTS database from the data portal of the Natural History Museum via https://data.nhm.ac.uk/dataset/the-2016-release-of-the-predicts-database-v1-1. The source data necessary to reproduce all the figures in this study, can be found in the following repository: https://figshare.com/articles/dataset/CarbonFoodNature_TradeOffs_-_Source_data/29618120.

## Code availability
The code used to generate the results is freely accessible and available at https://github.com/sarahgall/CarbonFoodNature_TradeOffs. The code used for the species distribution model was obtained from the JNCC SDM ensemble and can be downloaded via https://github.com/jncc/sdms.

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

## Acknowledgements

This work is supported in part by funds from the Engineering and Physical Sciences Research Council (EPSRC) and the Livestock, Environment and People project of the Oxford Martin School funded by the Wellcome Trust.

## Author contributions

S.S.G. conceptualised the study, collected the data, designed, implemented and ran the model, analysed the results, and wrote the paper; T.H. supplied the habitat condition data and advised the biodiversity modelling; M.O. gave advice about the agricultural and forestry model and gave advice about the manuscript; J.W.H. proposed the initial idea, supervised the project and edited the manuscript.

## Competing interests

The authors declare no competing interests.
