## [Transparent Peer Review file · Communications Earth & Environment]

Strategic land reallocation enhances carbon sequestration and biodiversity protection without compromising agricultural productivity in Great Britain

Corresponding Author: Ms Sarah Gall

Version 0:

Decision Letter:

Dear Ms Gall,

Your manuscript titled "Mapping the option space for carbon sequestration, food and biodiversity in Great Britain" has now been seen by 3 reviewers, and we include their comments at the end of this message. They find your work of interest, but some important points are raised. We are interested in the possibility of publishing your study in Communications Earth & Environment, but would like to consider your responses to these concerns and assess a revised manuscript before we make a final decision on publication.

We therefore invite you to revise and resubmit your manuscript, along with a point-by-point response that takes into account the points raised. In particular, for publication in Communications Earth & Environment we request that you:

*****Provide novel and fully supported insight into the land-use trade-offs between carbon sequestration, biodiversity, and agriculture/forestry in Great Britain.**

*****Outline your method in detail, including the multi-objective optimization procedure, explain and justify the choice of the step size of five for weighting, clarify potential sampling bias, and include all assumptions on carbon sequestration.**

*****Expand your discussion to economic cost, account for policy implications, further outline the limitation of your approach and your findings and demonstrate that your data and analysis fully support all your claims.**

Please submit your point-by-point responses as a separate file, distinct from your cover letter where you can add responses to the Editors' comments that you do not want to be made available to the reviewers. Word files are preferred. We recommend that any figures, tables or graphs that are included in the response to reviewers are also included in the main article or Supplementary Information.

Please use the following link to submit your revised manuscript, point-by-point response to the referees' comments (which should be in a separate document to any cover letter), a tracked-changes version of the manuscript (as a PDF file) and the completed checklist:

Link Redacted

We hope to receive your revised paper within six weeks; please let us know if you aren't able to submit it within this time so that we can discuss how best to proceed. If we don't hear from you, and the revision process takes significantly longer, we may close your file. In this event, we will still be happy to reconsider your paper at a later date, as long as nothing similar has been accepted for publication at Communications Earth & Environment or published elsewhere in the meantime.

Please do not hesitate to contact us if you have any questions or would like to discuss these revisions further. We look forward to seeing the revised manuscript and thank you for the opportunity to review your work.

Best regards,

Qiming Zheng, PhD
Editorial Board Member
Communications Earth & Environment
orcid.org/0000-0002-7393-6585

Martina Grecequet, PhD
Senior Editor
Communications Earth & Environment

EDITORIAL POLICIES AND FORMATTING

Editorial Policy: [Policy requirements](https://www.nature.com/documents/nr-editorial-policy-checklist.pdf) (Download the link to your computer as a PDF.)

- Behavioural and social science
- Ecological, evolutionary & environmental sciences
- Life sciences

<https://www.nature.com/documents/nr-reporting-summary.zip>

Furthermore, please align your manuscript with our format requirements, which are summarized on the following checklist: [Communications Earth & Environment formatting checklist](https://www.nature.com/documents/commsj-phys-style-formatting-checklist-article.pdf)

and also in our style and formatting guide [Communications Earth & Environment formatting guide](https://www.nature.com/documents/commsj-phys-style-formatting-guide-accept.pdf) .

*** DATA: Communications Earth & Environment endorses the principles of the Enabling FAIR data project (<http://www.copdess.org/enabling-fair-data-project/>). We ask authors to make the data that support their conclusions available in permanent, publically accessible data repositories. (Please contact the editor if you are unable to make your data available).

All Communications Earth & Environment manuscripts must include a section titled "Data Availability" at the end of the Methods section or main text (if no Methods). More information on this policy, is available at <http://www.nature.com/authors/policies/data/data-availability-statements-data-citations.pdf>.

If a community resource is unavailable, data can be submitted to generalist repositories such as [figshare](https://figshare.com/) or [Dryad Digital Repository](http://datadryad.org/). Please provide a unique identifier for the data (for example a DOI or a permanent URL) in the data availability statement, if possible. If the repository does not provide identifiers, we encourage authors to supply the search terms that will return the data. For data that have been obtained from publically available sources, please provide a URL and the specific data product name in the data availability statement. Data with a DOI should be further cited in the methods reference section.

REVIEWER COMMENTS:

Reviewer #1 (Remarks to the Author):

This paper presents a spatially explicit multi-objective optimization approach to assess land-use trade-offs between carbon sequestration, biodiversity, and agriculture/forestry in Great Britain. By using Pareto frontiers and scenario analysis, the study evaluates trade-offs between different objectives and introduces a land conversion budget, which helps explore the entire decision space while increasing the flexibility and practical applicability of the results in policymaking. However, the paper seems to lack in-depth discussion of the research findings, and the methodology lacks some degree of rigor. Therefore, I think the paper requires major revisions, and the suggestions are as followed:

1. When calculating biodiversity metrics, the NBN atlas database may be subject to sampling bias, particularly in remote areas where data may be sparse, potentially leading to an underestimation of biodiversity potential in those regions. How does the study account for or mitigate this potential bias?
2. In the optimization analysis, the impact of input data uncertainty (e.g. carbon sequestration, production, biodiversity) on optimization results has not been discussed.
3. Why was a step size of 5 chosen? Would different step sizes or different weight distributions lead to different optimal land-use allocations?
4. I suggest adding a discussion on the economic costs of land conversion, time delays, and other factors that may affect the feasibility of implementing the optimization results.
5. For Figure 1, it would be helpful to include a quantitative description of the contributions of different land-use conversions to the overall optimization outcomes.
6. The paper investigates the optimization of objectives with a resolution of 500m x 500m, but some of the input datasets seem to have inconsistent resolutions. How should this issue of resolution mismatches be addressed?
7. In the land conversion budget results section, could the paper provide insights into how the optimal conversion areas change under different budget constraints?
8. The hotspots for change lack sufficient explanation—additional discussion may improve clarity.
9. The economic costs of land conversion are not mentioned in the study; this should be acknowledged in the limitations section.
10. How do the study's results compare to other similar studies conducted in GB? A comparative analysis would strengthen the discussion.
11. In the calculation of carbon sequestration for plantation forests, was the impact of harvesting considered? If not, this could lead to an overestimation of sequestration potential.
12. The study assumes that current environmental conditions remain constant, but I think this may lead to simplified results with some bias and limitations. I think that climate modelling scenarios should be introduced or just qualitative analyses under different climate change scenarios may also add some value.

Reviewer #2 (Remarks to the Author):

The authors work is meaningful, while may need more clarification to allow more readers understand it easily. I have a few suggestions on achieving that:

Line 20: Both "climate mitigation" and "environmental conservation" are vague. What did you refer to?

Line 83-128: The authors introduced different kinds of existing land use models and their pros and cons, respectively. However, it lacks logical connection with the previous paragraph on GB, i.e., what is the major problem in GB and which kind of models are more suitable for solving the problem? Meanwhile, how does the model in this paper outcompete the limitations in previous models? I am wondering if a table summarizing the existing models and comparing them with the model used in this study will help the readers understand the status. Or a conceptual framework addressing the current land use issues in GB and how different models will solve the issues based on their different emphasis or pros.

Line 147: The results should stand on its own. You may consider a brief introduction on how to quantify the performance, what are the objective you focus here, and why the values in Table 1 varies so much across different objectives

Line 154: why 35-35-30 instead of 33-33-33? Or will it be large difference between these two? Additionally, your normalization seems 0-1 in Fig. 1, why weighting combinations here is 0-100? Are these numbers percentages?

Line 190-213: I was initially wondering what the changes in the performance values means in real conservation efforts, until I kind of found the answer in the next paragraph (line 214). It would be better to make it clear what changed in the performance represent for each objectives, e.g., change in 0.1 of performance in production represent xx billion £-yr⁻¹, or change in xx million t CO₂-eq-yr⁻¹ carbon sequestration, if my understanding is correct. Additionally, the author may also need to explain the chosen of unit in each objective. I realized the details could be in the method section, while given the current structure of the paper, I prefer it be readable without checking the method since method is in the end of the paper.

Line 223: would be good to explain what "biodiversity score" means or represent

Line 285: provide more details on the setting of budget gradient, how it was defined and measured, why it is in percentage, what it means in real management, 1% vs. 100%?

Line 300: consider adjusting the figure and the color gradient. I can not tell 10% from 20%, and whether the shape is linear or convex.

Line 311: are there any turning points of budget for that?

Line 695: what are the 88 species and are they representative enough?

Line 709: how was the calculations are actually done? The reader may need to understand this without checking through another published paper.

Reviewer #3 (Remarks to the Author):

This manuscript addresses a critical topic concerning land-use optimization for Great Britain, providing valuable insights into balancing carbon sequestration, agricultural productivity, and biodiversity conservation. The authors utilize robust multi-objective optimization methodologies, present novel high-resolution analyses, and demonstrate clear implications for national policy-making. However, substantial revisions are needed to enhance readability, better justify methodological choices, and strengthen the discussion on policy implications and limitations.

The manuscript, while methodologically sound, suffers from overly complex descriptions in parts, lacks sufficient justification for some key assumptions, and would benefit significantly from more refined visualizations and clearer interpretation of the results. A revised version must address these clarity and justification issues explicitly.

1. Lines 18-31: Clearly state your novel contribution to distinguish your research from other existing work explicitly in the abstract.
2. Lines 69-82: Clarify explicitly why leaving the EU's Common Agricultural Policy provides a unique and timely opportunity for your research, linking this context more strongly to your objectives.
3. Lines 111-122: More clearly articulate the gap in existing modeling literature that your spatially explicit analysis addresses.
4. Lines 152-154: Explain explicitly why you chose the step size of five for the weighting discretization and justify why alternative step sizes were not considered.
5. Lines 186-207: The detailed numeric descriptions (percentages and values) in-text are overly complex. Consider summarizing these more concisely, shifting detailed numeric descriptions into supplementary materials or tables.
6. Lines 226-264: Explain explicitly why converting semi-natural habitats to pasture is beneficial in your model since such conversions typically imply biodiversity losses. Clarify this seemingly counterintuitive result.
7. Lines 437-465: Provide a more explicit and structured discussion on practical challenges (e.g., political feasibility, socio-economic impacts, cultural acceptance) associated with the suggested land-use transitions.
8. Lines 466-480: Discuss how different assumptions around carbon sequestration in timber products might alter policy recommendations, given their significant role in your findings.
9. Lines 481-506: Expand your discussion on the potential impacts on food security, particularly addressing the trade-offs involved if dietary changes are not realized.
10. Lines 507-522: Clearly discuss the potential limitations arising from using the selected biodiversity indicators. Highlight how including habitat connectivity might change your recommendations.
11. Lines 523-539: Clarify explicitly the implications of not considering urban expansion in your model and how future urban development pressures might affect the feasibility of your land-use optimization.
12. Lines 580-614: Clarify the rationale behind not excluding protected areas explicitly from your potential land conversions. Discuss potential implications or biases that might result from this decision.
13. Lines 615-657: Provide additional details on the assumptions underpinning your carbon sequestration calculations, explicitly acknowledging the uncertainties related to emissions from pastures and plantations.
14. Lines 658-683: Clarify explicitly why and how you chose the species used for the biodiversity modeling. Discuss any limitations or potential biases inherent in these selections.
15. Lines 684-724: Clearly outline the multi-objective optimization procedure step-by-step, providing a simplified illustrative example that helps readers unfamiliar with such methods better understand your analytical process.

Communications Earth & Environment is committed to improving transparency in authorship. As part of our efforts in this direction, we are now requesting that all authors identified as 'corresponding author' create and link their Open Researcher and Contributor Identifier (ORCID) with their account on the Manuscript Tracking System prior to acceptance. ORCID helps the scientific community achieve unambiguous attribution of all scholarly contributions. You can create and link your ORCID from the home page of the Manuscript Tracking System by clicking on 'Modify my Springer Nature account' and following the instructions in the link below. Please also inform all co-authors that they can add their ORCIDs to their accounts and that they must do so prior to acceptance.

If you experience problems in linking your ORCID, please contact the [Platform](http://platformsupport.nature.com/)

Support Helpdesk.

Version 1:

Decision Letter:

Dear Ms Gall,

Your manuscript titled "Mapping the option space for carbon sequestration, food and biodiversity in Great Britain" has now been seen by our reviewers, whose comments appear below. In light of their advice we are delighted to say that we are happy, in principle, to publish a suitably revised version in Communications Earth & Environment.

We therefore invite you to revise your paper one last time to address the remaining concerns of our reviewers. At the same time we ask that you edit your manuscript to comply with our format requirements and to maximise the accessibility and therefore the impact of your work.

EDITORIAL REQUESTS:

*****Please take care to match our formatting and policy requirements. We will check revised manuscript and return manuscripts that do not comply. Such requests will lead to delays. *****

SUBMISSION INFORMATION:

OPEN ACCESS:

Communications Earth & Environment is a fully open access journal. Articles are made freely accessible on publication. For further information about article processing charges, open access funding, and advice and support from Nature Portfolio, please visit <https://www.nature.com/commsenv/open-access>

Link Redacted

Best regards,

Qiming Zheng, PhD
Editorial Board Member
Communications Earth & Environment

Martina Grecequet, PhD
Senior Editor,
Communications Earth & Environment
Consulting Editor,
Communications Sustainability

REVIEWERS' COMMENTS:

Reviewer #1 (Remarks to the Author):

The author has provided a thorough and clear response to the issues raised, and has made specific revisions to the manuscript, including additions to the analysis and discussion of the results. I think my concerns have been fully addressed, including the justification of the step size and the limitations regarding land conversion feasibility. The additional figures have also improved the readability, and the revised version represents a significant improvement over the original submission. However, I recommend that the author carefully review the use of certain terms in the final manuscript, such as distinguishing between "carbon storage" and "carbon sequestration".

Reviewer #2 (Remarks to the Author):

The authors have done a sufficient amount of revisions in response to my previous comments and I appreciate their efforts on improving the ms. I find all the responses satisfying and do not have more comments.

Reviewer #3 (Remarks to the Author):

The manuscript is well organised and technically sound; it already meets most standards for publication. Only minor textual corrections and consistency edits are needed. No additional analyses are required.

1. Abbreviations on first use – e.g. "UK" is introduced early but "GB" appears later; confirm both are defined on first use.
2. Units and symbols – Ensure a non-breaking space between number and unit (e.g. "500 m", "25 m raster"), and italicise variables in equations.
3. Capitalisation of headings – The journal's style guide may prefer either sentence case or title case; align section and subsection headings accordingly.

** Visit Nature Portfolio's author and referees' website at www.nature.com/authors for information about policies, services and author benefits**

RESPONSES TO REVIEWER COMMENTS

Reviewer 1:

This paper presents a spatially explicit multi-objective optimization approach to assess land-use trade-offs between carbon sequestration, biodiversity, and agriculture/forestry in Great Britain. By using Pareto frontiers and scenario analysis, the study evaluates trade-offs between different objectives and introduces a land conversion budget, which helps explore the entire decision space while increasing the flexibility and practical applicability of the results in policymaking. However, the paper seems to lack in-depth discussion of the research findings, and the methodology lacks some degree of rigor. Therefore, I think the paper requires major revisions, and the suggestions are as followed:

Reviewer 1 - Comment 1:

When calculating biodiversity metrics, the NBN atlas database may be subject to sampling bias, particularly in remote areas where data may be sparse, potentially leading to an underestimation of biodiversity potential in those regions. How does the study account for or mitigate this potential bias?

Response:

Thank you for your comment. We acknowledge that the NBN Atlas database, like many biodiversity datasets, may be subject to sampling bias, particularly in remote or less-studied regions.

We did not correct for sampling bias but instead based our species selection on filtering from an existing big land-use model called NEVO. For the NEVO model, one of the selection criteria was sampling sufficiency in the form of records collected in the last 20 years.

Additionally, we have stated this potential limitation in the method section in Lines 832- 837 (Lines 848 – 851 in tracked changes version) and have now included this in the discussion in Lines 611 – 612 (Lines 622 - 623 in tracked changes version).

Revised text in Lines 832- 837 (Lines 848 – 851 in tracked changes version):

“This selection was based on a list of priority species that have also been used in the SDM run as part of the NEVO land-use model⁴² and was composed based on a range of criteria including the number of records in the last 20 years, the spreading of a species and independence from water quality⁹⁸. The species occurrence data was downloaded from the NBN atlas database⁹⁹, which has some limitations due to it not being systematically surveyed but is the most comprehensive dataset available.”

And in Lines 611 – 612 (Lines 622 - 623 in tracked changes version):

“Uncertainty in the input data includes a potential sampling bias in the biodiversity occurrence data”

Reviewer 2 – Comment 2:

In the optimization analysis, the impact of input data uncertainty (e.g. carbon sequestration, production, biodiversity) on optimization results has not been discussed.

Response:

This is indeed a critical point and we are happy to respond: While preparing the input data, we sense checked and compared different datasets to make sure the most suitable data was chosen.

Nevertheless, we agree that uncertainty in input data may influence the optimisation results, and readers need to be made aware. To address the issue, we have now expanded our discussion in Lines 610 – 622 (Lines 621-635 in the tracked changes version) to consider the potential impact of input data uncertainty on our results.

This is the revised text in Lines 610 – 622 (Lines 621-635 in the tracked changes version):

*“As for most models, the results depend on the data input for the benefit calculations. **Uncertainty in the input data includes a potential sampling bias in the biodiversity occurrence data, a lack of including differences in soil composition beyond peaty soils into soil carbon storage, potential spatial differences in forest age composition. Additionally, data for forest growth and agricultural yields is based on modelling and might include uncertainties about the realised local yields especially under locally adapted management practices. Wherever possible, national datasets were used to increase the accuracy; no such dataset was available for arable yields. Thus, a global dataset had to be used instead. For the same reason, the spatially explicit woodland species composition data comes from a Europe-wide dataset. Therefore, our results should not be used to prescribe what exactly should happen in a specific location but rather be interpreted as a general exploration of spatial trade-offs in GB. Additionally, benefit estimates can differ depending on the chosen indicators. Especially for biodiversity, using different indicators can lead to differing results.**”*

Reviewer 1 – Comment 3: Why was a step size of 5 chosen? Would different step sizes or different weight distributions lead to different optimal land-use allocations?

Response:

We acknowledge that this point needs further clarification in our manuscript. We have now added a more detailed explanation for the choice of a step size of five in the weighting discretization. This step size was selected to balance the necessary number of scenarios and therefore their comprehensibility, computational efficiency, and resolution. A finer step size (e.g., one or two) would have significantly increased the number of scenarios generated, (with:

- stepsize 10 -> 66 scenarios;
 - stepsize 5 -> 231 scenarios;
 - stepsize 2 -> 1326 scenarios;
 - stepsize 1 -> 5151 scenarios)
- and the computational complexity without substantial improvements in the results, while a larger step size (e.g., ten) would have led to excessive discretisation, which does not allow the creation of a smooth pareto frontier. The land allocation also does not change significantly when a smaller step size is chosen, as you can see in the plots we added to the SI in Figures S1 – S4.

To clarify our choice, we have updated the manuscript in the Results section (Lines 169 – 171 and Lines 171 – 173 in the tracked changes version) and the Methods section (Lines 883 – 889 and Lines 901 – 907 in the tracked changes version). Additionally, we have added the resulting plot of the frontiers and the land allocation for a step size of 10 and 2 in the Appendix, to show what the effect would be.

Revised Text in Lines 169 – 171 (Lines 171 – 173 in the tracked changes version):

“The step size was chosen based on some preliminary tests to balance the number of scenarios and the resolution needed to create a smooth pareto frontier (see Figure S1 – Figure S4).”

And in Lines 883 – 889 (Lines 901 – 907 in the tracked changes version):

“The step size of five was chosen as a trade-off between resolution and the comprehensibility of the number of scenarios, as well as the computational runtime. A smaller step size would have increased the number of scenarios significantly, which makes exploring them infeasible and increases the computational runtime, while a larger step size would have led to excessive discretisation and made it impossible to draw the full pareto-frontier (see Figure S1 – Figure S4). Preliminary tests showed that a step size of five was able to create a smooth pareto frontier while keeping the number of scenarios manageable.”

Reviewer 1 – Comment 4: I suggest adding a discussion on the economic costs of land conversion, time delays, and other factors that may affect the feasibility of implementing the optimization results.

Response: Thank you for making this important point. We acknowledge that while the optimisation framework provides a useful theoretical guide for prioritising land-use change, there are several real-world constraints that might affect its implementation. These include the economic costs of land conversion, the time needed for new ecosystems to establish, or for policy implementation. We have added this to our discussion of real-world constraints in 593 – 609 (Lines 604 – 620 in the tracked changes version) and highlighted them as potential avenues for future research.

Revised text in Lines 593 – 609 (Lines 604 – 620 in tracked changes version):

“Additionally, other factors might affect the feasibility of implementing the results of our analysis, such as land ownership or local political context and capacities. The economic costs of land conversion are another constraint on implementation that has not been included in this analysis. While the agricultural production calculated for this study gives an idea about where opportunity costs for farmers are higher in the case of conversion to other area, it does not include the actual cost of new policy schemes for incentivising landowners, the cost of conversion to different land-use types or other cost that might occur in the implementation. For many land conversions, private costs exceed private benefits, and public funding will be necessary to enable these measures²¹. For instance, for planting broadleaved forests, the CCC estimated a net present value of -25,600 per hectare, including the costs for land purchasing, financing, planting and establishment²¹. Higher conversion rates might be less feasible when these constraints are included. Additionally, time delays from the time it takes for land-uses and ecosystems to establish and deliver their full benefits have not been considered in this study and might affect the real-life outcomes from land conversions. Future work could include these aspects to investigate which conversions are not only beneficial but also feasible when including factors such as land ownership and economic costs of land conversion.”

Reviewer 1 – Comment 5: For Figure 1, it would be helpful to include a quantitative description of the contributions of different land-use conversions to the overall optimization outcomes.

Response:

We are thankful for this suggestion as it helps improving the clarity of the paper. We have now added a brief paragraph to the results section describing the conversion patterns that contribute to the optimisation results. The main challenge here was that these would look different for each of the 231 scenarios, while it is infeasible to describe all of them. That’s why we chose scenario aggregations like in the “Current land-use and strictly better scenarios”-Section, the “Identifying hotspots for change”-Section and the “Priority grouping analysis”-Section, where a broader range of scenarios is summarised. The “Current land-use and strictly better scenarios”-Section

shows what the strictly better scenarios (in the upper right quadrant from the current scenario) have in common, and the “Priority grouping analysis”-Section summarises the scenarios in four groups (carbon prioritising, production prioritising, biodiversity prioritising and balances).

Nevertheless, we agree that an overview of the conversion patterns that contribute to the optimisation results would enhance our “Pareto-optimal land-uses” Section and the interpretability of Figure 1. Therefore, we have now added a brief paragraph in Lines 209-223 (Lines 211-225 in tracked changes version) that describes these patterns as well as Table S2 in the SI, which shows the land-use composition in all 231 scenarios and Figure S5 with example maps to show the spatial differences scenarios with different priority weightings.

The revised text in Lines 209 – 223 (Lines 211 – 225 in the tracked changes version):

“Looking at the land conversions that correspond to the points along the production-carbon pareto frontier in Figure 1a, frequent conversions from pasture to arable land and semi-natural habitats to pasture and some conversions from semi-natural habitat to plantation forests can be seen in the scenarios scoring high on production. However, the scenarios with higher values for carbon sequestration show higher conversion rates from arable land and pasture to semi-natural habitat. The 2D pareto-frontier for production and species occurrence (see Figure 1b) shows very similar conversions for the more production-focused scenarios. The scenarios with higher biodiversity performances show high conversion rates from pasture and arable to semi-natural habitats and some conversions from plantation forests to semi-natural habitats. For the very short frontier in Figure 1c the conversions seen in the pareto-efficient scenarios are very similar, with high conversion rates from arable land and pasture to semi-natural habitats. The main difference between the scenarios is that the ones scoring higher on biodiversity show even higher conversion rates to semi-natural habitat from all land-use types, and they include conversions from plantation forests to semi-natural habitat, which are rarer in the more carbon focused scenarios. A table that shows the land conversion shares for all 231 scenarios can be found in the SI (Error! Reference source not found.). “

And the new Figure S5:

Reviewer 1 – Comment 6: The paper investigates the optimization of objectives with a resolution of 500m x 500m, but some of the input datasets seem to have inconsistent resolutions. How should this issue of resolution mismatches be addressed?

Response:

Thank you for pointing out the potential issue with resolution mismatches. We agree that inconsistencies in spatial resolution between the 500m x 500m optimization grid and the input datasets can affect the accuracy and interpretation of the results. To address this, we have taken the following steps:

- All datasets have been reprojected to the same resolution, using a resampling technique suitable for the data category
- Most datasets with differing resolutions to the 500m grid have coarser resolution, such as 1km or more. Those were resampled using either a nearest neighbour algorithm or a bilinear sampling method
- For the only dataset where the resolution had to be resampled to a coarser resolution (the CEH land-use map), we analyzed what resampling method shows the best outcomes regarding the distribution of land-use types being consistent with the original map. To show this, we have added Figure S7 in the SI
- Additionally, we now explicitly acknowledge the potential influence of resolution mismatches in the revised manuscript and discuss what resampling methods were chosen to align the resolutions of the datasets

Revised text in the Methodology in Line 722:

"The Land Cover Map 2020 (25m rasterised land parcels, GB) published by UKCEH⁵⁹ was aggregated to a resolution of 500m using a nearest-neighbour resampling approach, which allowed to maintain the original shares of land-use classes (see Figure S7)"

Additionally, in Lines 733-738 (Lines 749-754 in the tracked changes version) we added the following text:

"To ensure consistency, all input datasets were resampled to the 500m x 500m resolution of the optimization grid using a nearest-neighbour or bilinear resampling algorithm. We also conducted a sensitivity analysis to evaluate the impact of different resampling strategies on the input data. While resolution mismatches were minimised, we acknowledge that some residual effects may remain and could influence outcomes in areas with high spatial variability."

And Figure S7:

Reviewer 1 – Comment 7: In the land conversion budget results section, could the paper provide insights into how the optimal conversion areas change under different budget constraints?

Response:

Thank you for this suggestion. We agree that providing additional insights into how **optimal conversion areas shift under different budget constraints** would enhance the clarity of our results. The main challenge here was to give an idea of the overall patterns without describing an unfeasible number of scenario weightings throughout several budgets. Therefore, we de-

cided to show four example scenarios (one with a slight prioritisation of one of the three objectives, and a balanced one) for a range of budgets as plots in the SI. Additionally, we have expanded our ‘Conversion budgets’ results section in Lines 322-335 (Lines 332 – 342 in the tracked changes version) to briefly describe how the land conversion patterns develop with increasing budgets.

Revised text in Lines 322-335 (Lines 332 – 342 in the tracked changes version):

*“For each combination of weights, we identify the ranked list of conversions across the entire country, based on the weighted sum of their performance. We then identify how many of those ranked conversions can be implemented within the total ‘budget’ of permitted conversions. **Cells with the most prominent overall benefits under a weighting combination are chosen first, while less beneficial cells are added in merit order with higher budgets.** The analysis was done for a range of conversion budgets, representing the share of convertible land that is made available for conversion. This enables decision makers to explore how much change would be necessary to achieve certain outcomes and therefore how ambitiously land conversion targets need to be set. To explore a broad range of conversion rates, 11 budgets were analysed, starting with a budget of 1% and then stepping in increments of 10%, up to 100% of the rural land available for conversion. The resulting pairwise-pareto frontiers for the budgets in intervals of 10% are shown in **Error! Reference source not found.** In **Figure S6 in the SI**, the spatial conversion patterns of four example scenarios (one with a prioritisation of each of the three objectives respectively and one with a balanced weighting prioritisation) are shown for increasing budgets.”*

And the new Figure S6:

Reviewer 1 – Comment 8: The hotspots for change lack sufficient explanation—additional discussion may improve clarity.

Response:

Thank you for spotting this insufficiency. We have now expanded our explanation in Lines 384-392 (Lines 392 – 401 in the tracked changes version) in the ‘Identifying hotspots for change’ section in the Results to make it clearer. Specifically, we have explained in more detail how the hotspot for change maps were created, what we mean when we say that they show conversions that are more robust, and what exactly Figure 4 is showing.

The revised text in Lines 384-392 (Lines 392 – 401 in the tracked changes version):

*“To identify priority areas for land conversion, the scenarios **with 231 priority weightings are combined by calculating the frequency of conversion in each cell across all scenarios. This allows us to generate a ‘heatmap’ of conversion frequencies. Cells converted under a broader range of weighting combinations can be considered ‘hotspots for change’, which are beneficial while showing comparatively smaller trade-offs. The more often a conversion occurs throughout the scenarios, the more priority weighting combinations it satisfies. It is therefore more robust to changing priorities. Figure 4 shows the relative frequency of change in each cell and the frequency of change to each land-use category (arable land, pasture, plantation forest, and semi-natural habitats), expressed as a percentage share of the 231 scenarios.**”*

Reviewer 1 – Comment 9: The economic costs of land conversion are not mentioned in the study; this should be acknowledged in the limitations section.

Response:

We agree that the economic costs of land conversion will be essential considerations at the stage of implementation. Our study, which of exploratory character, aims at studying trade-offs measured along (bio-)physical dimensions using a weighted optimisation approach rather than financial optimisation using a unifying currency. Thus, we have designed this study to focus on biophysical and ecological conditions and have not included the financial costs of conversions in the optimisation. We have now acknowledged this limitation in the Discussion in Lines 593-609 (Lines 604 – 620 in tracked changes version), together with the discussion of discussion on the economic costs of land conversion, time delays, and other factors that may affect the feasibility of implementing the optimisation results as mentioned in Comment 4.

This is the revised text in Lines 593-609 (Lines 604 – 620 in the tracked changes version):

*“**Additionally, other factors might affect the feasibility of implementing the results of our analysis, such as land ownership or local political context and capacities. The economic costs of land conversion are another constraint on implementation that has not been included in this analysis. While the agricultural production calculated for this study gives an idea about where opportunity costs for farmers are higher in the case of conversion to other area, it does not include the actual cost of new policy schemes for incentivising landowners, the cost of conversion to different land-use types or other cost that might occur in the implementation. For many land conversions, private costs exceed private benefits, and public funding will be necessary to enable these measures²¹. For instance, for planting broadleaved forests, the CCC estimated a net present value of -25,600 per hectare, including the costs for land purchasing, financing, planting and establishment²¹. Higher conversion rates might be less feasible when these constraints are included. Additionally, time delays from the time it takes for land-uses and ecosystems to establish and deliver their full benefits have not been considered in this study and might affect the real-life outcomes from land conversions. Future work could include these aspects to investigate which conversions are not only beneficial but also feasible when including factors such as land ownership and economic costs of land conversion.**”*

Reviewer 1 – Comment 10: How do the study’s results compare to other similar studies conducted in GB? A comparative analysis would strengthen the discussion.

Response: Thanks for your suggestion. We have added to the Discussion to include a comparison with other land-use studies that were carried out for GB.

This is the revised text in Lines 536-546 (Lines 547 – 557 in the tracked changes version):

“These findings are in line with other UK-focused land-use studies. Smith (2022) concludes in a non-spatial study that the pressure on the land is high when trying to achieve several sustainability objectives while ensuring food security, but that policies that encourage more sustainable land-use could turn the land-use sector into a carbon sink³⁴. Similarly, the CCC states that a reduction of land-based emissions is possible while achieving other objectives, such as food production and biodiversity, but will require radical changes in land-use³². While both studies are not spatially explicit, they additionally include changes in consumption behaviour and management practices in their calculations, which have not been included in this study. The strong synergies we find for planting of natural broadleaved forests in the right locations match the call for careful design of afforestation policies that should focus on creating natural forests instead of biodiversity-poor conifer monocultures⁵³ and in the right locations⁵⁴.”

Reviewer 1 – Comment 11: In the calculation of carbon sequestration for plantation forests, was the impact of harvesting considered? If not, this could lead to an overestimation of sequestration potential.

Response:

Thank you for raising this and we appreciate the opportunity to clarify this point. The impact of harvesting on carbon sequestration in plantation forests was explicitly considered in our analysis, in Lines 784-790 (Lines 805-806). Specifically, we multiply the amount of carbon sequestered in plantations by the share of long-term storage in harvested wood products. We assume that only 48% of carbon sequestered is stored long-term; this number is based on recreating the harvested wood product calculations by Peng et al. (2023) for the UK, based on wood product statistics for the UK.

To ensure this information is accessible and clearly communicated, we have added a sentence to explicitly highlight this consideration in the Methodology section in Lines 784-790 (Lines 805-806 in the tracked changes version).

Additionally, our assumptions for carbon sequestered by managed plantation forests include thinning and wood removal as part of the forest management, based on the numbers in the woodland carbon code look-up tables. This is explained in Table S5, which gives an overview of all assumptions on forest management and forest growth in the carbon sequestration model.

This is the revised text in Lines 784-790 (Lines 805-806 in the tracked changes version):

“To estimate the ratio of long-term carbon storage in harvested wood products, the global wood product model by Peng et al. (2023)⁵² was recreated for the UK using the FAOSTAT Forestry Production and Trade data for the UK in 2022⁷⁸ and the Forest Research Forestry Statistics 2023^{79,80}. Average yearly numbers for carbon sequestration/emissions from plantation woodlands and natural broadleaved forests calculated over the period 2020-2050. For plantation woodlands, this number is then multiplied by the share of long-term carbon storage in harvested wood products.”

Reviewer 1 – Comment 12: The study assumes that current environmental conditions remain constant, but I think this may lead to simplified results with some bias and limitations. I think that climate modelling scenarios should be introduced or just qualitative analyses under different climate change scenarios may also add some value.

Response:

Thank you for this valuable comment. We agree that assuming static environmental conditions is a simplification that may influence the applicability of the results under future climate change. In this study, we intentionally focused on current conditions to clearly establish and communicate the optimisation framework and trade-offs among the considered objectives.

However, we fully acknowledge the importance of considering climate change and its potential impacts on arable and forest yields, biodiversity, etc. To address this, we are currently developing a follow-up study that explicitly integrates multiple climate change scenarios into the model. We have now clarified this point in the revised Discussion, in Lines 654-657 (Lines 669 – 673 in tracked changes version), and have added a sentence about the ongoing work that will explore the effect of climate change on land-use trade-offs.

Revised text: in Lines 654-657 (Lines 669 – 673 in tracked changes version):

*“Additionally, land suitability could change under future climate change scenarios, especially for agricultural production⁵⁷. These potential future changes have not been included in this static analysis, **but subsequent research will incorporate future climate scenarios into the modelling framework to enable more long-term planning under changing climatic conditions.**”*

Reviewer 2:

The authors work is meaningful, while may need more clarification to allow more readers understand it easily. I have a few suggestions on achieving that:

Reviewer 2 – Comment 1: Line 20: Both “climate mitigation” and “environmental conservation” are vague. What did you refer to?

Response:

We agree that these terms could be more specific. In the revised manuscript, we have changed the wording of the abstract to clarify what aspects of climate mitigation and environmental conservation we are referring to. Specifically, we replaced the term “climate mitigation” with “carbon emissions and sequestration” and the term “ environmental conservation” with “biodiversity”.

This is the revised text in Lines 19 - 21:

*“Due to the various negative environmental consequences of current land-use, and land’s important role **regarding carbon emissions and sequestration, biodiversity and food security,** there is a growing and urgent interest in reforming land-use in many countries.”*

Reviewer 2 – Comment 2: Line 83-128: The authors introduced different kinds of existing land-use models and their pros and cons, respectively. However, it lacks logical connection with the previous paragraph on GB, i.e., what is the major problem in GB and which kind of models are more suitable for solving the problem? Meanwhile, **how does the model in this paper outcompete the limitations in previous models?** I am wondering if **a table summarizing the existing models** and comparing them

with the model used in this study will help the readers understand the status. Or a conceptual framework addressing the current land-use issues in GB and how different models will solve the issues based on their different emphasis or pros.

Response:

Thanks for raising this point as it helped us making our contribution more explicitly and precise. We agree that we can make the connection between the policy context in GB and the models with their advantages and disadvantages clearer. In addition, we want to highlight that we did not try to make the point that our model outcompetes all other existing models, but rather that each type of model is designed to evaluate a certain aspect of land-use policy making and answer a certain set of policy-related questions. Different challenges need different models to be addressed. The existing large-scale land-use models were designed to answer research questions different from exploring best-case scenarios dependent on priority weighting and the analysis of trade-offs between a set of objectives. To make this clearer, we have adjusted the Introduction in Lines 85-89 and Lines 115-119.

Additionally, as suggested by your comment, we have added a Table (Table S1) to the SI that compares the advantages and disadvantages of certain models and what kind of policy questions they can answer.

The revised text in lines 85-89 (Lines 86-90 in the tracked changes version):

“Land-use modelling can help deliver the evidence needed to design new strategies and policies. There is a range of existing models and approaches for modelling land-use, **with each type of model being suitable for answering a certain set of policy-relevant questions. A detailed overview of existing land-use models and the policy-relevant questions they can answer can be found in Table S1.** A common approach is the use of ...”

And in lines 115-119 (Lines 116 – 121 in tracked changes version):

“While all these models help answer important aspects related to the design of new land-use policies, none of them is suitable for evaluating the land-use interdependencies between different objectives from an explicitly spatial perspective. For this purpose, spatial trade-off modelling is more appropriate. This type of modelling is used in many contexts to identify optimal locations for specific land-uses and spatial trade-offs.”

And the table we added to the SI (Table S1):

Model type & description	Pros	Cons	Policy relevance & questions that can be answered	Comparison with our model
Scenario-based models (e.g. MARK, CAPSIM, GISC, etc.) These models are based on ex-ante scenario planning, which evaluates the consequences of these scenarios.	• Easy to apply for large scale • Easy to compare with a baseline scenario	• Not suitable for exploring trade-offs • Not suitable for exploring the consequences of policy changes	• Policy relevance: scenario-based models are suitable for exploring the consequences of policy changes • Questions: What are the consequences of a given policy? What are the consequences of a given scenario?	• Scenario-based models are not suitable for exploring trade-offs • Scenario-based models are not suitable for exploring the consequences of policy changes
Agent-based simulation models These models simulate the behaviour of individual agents and their interactions with each other and with the environment.	• Good for exploring the consequences of policy changes • Good for exploring the consequences of policy changes	• Not suitable for exploring trade-offs • Not suitable for exploring the consequences of policy changes	• Policy relevance: agent-based models are suitable for exploring the consequences of policy changes • Questions: What are the consequences of a given policy? What are the consequences of a given scenario?	• Agent-based simulation models are not suitable for exploring trade-offs • Agent-based simulation models are not suitable for exploring the consequences of policy changes
Multi-criteria decision analysis (MCDA) These models are based on the evaluation of different scenarios based on a set of criteria.	• Good for exploring the consequences of policy changes • Good for exploring the consequences of policy changes	• Not suitable for exploring trade-offs • Not suitable for exploring the consequences of policy changes	• Policy relevance: MCDA models are suitable for exploring the consequences of policy changes • Questions: What are the consequences of a given policy? What are the consequences of a given scenario?	• Multi-criteria decision analysis models are not suitable for exploring trade-offs • Multi-criteria decision analysis models are not suitable for exploring the consequences of policy changes
Linear programming models These models are based on the optimization of a linear objective function subject to linear equality and inequality constraints.	• Good for exploring the consequences of policy changes • Good for exploring the consequences of policy changes	• Not suitable for exploring trade-offs • Not suitable for exploring the consequences of policy changes	• Policy relevance: linear programming models are suitable for exploring the consequences of policy changes • Questions: What are the consequences of a given policy? What are the consequences of a given scenario?	• Linear programming models are not suitable for exploring trade-offs • Linear programming models are not suitable for exploring the consequences of policy changes
Stochastic simulation models These models are based on the simulation of random events and their consequences.	• Good for exploring the consequences of policy changes • Good for exploring the consequences of policy changes	• Not suitable for exploring trade-offs • Not suitable for exploring the consequences of policy changes	• Policy relevance: stochastic simulation models are suitable for exploring the consequences of policy changes • Questions: What are the consequences of a given policy? What are the consequences of a given scenario?	• Stochastic simulation models are not suitable for exploring trade-offs • Stochastic simulation models are not suitable for exploring the consequences of policy changes

Reviewer 2 – Comment 3: Line 147: The results should stand on its own. You may consider a brief introduction on how to quantify the performance, what are the objective you focus here, and why the values in Table 1 varies so much across different objectives.

Response:

Thank you for your suggestion. We recognise the importance of ensuring that the results section is self-contained and that readers can understand how the three objectives are quantified without having to go to the Methods section.

The variations in Table 1 are caused by the extreme scenarios which they represent, meaning maximising potential benefits for each of the three objectives with an unlimited amount of rural land conversion. Consequently, the maximum value for production would, for example, come from a scenario where all of GB is converted to either arable land, pasture or a few plantation forests.

To address these points, we have made the following edits:

1. We clearly reintroduced the three objectives, carbon sequestration, (food and timber) production and biodiversity, at the beginning of the results section
2. We now provide a short explanation of how the benefits are quantified at the beginning of the results section
3. We changed the title of Table1 to make it clearer what the values represent

Revised text in Lines 155-162 (Lines 157-164 in tracked changes version):

“For each grid cell, we identified the land-use conversions that maximise and minimise each of the three objectives: Carbon sequestration, (food and timber) production and biodiversity. The performance of a land allocation scenario in terms of carbon sequestration consists of carbon fluxes from land conversion, emissions from agriculture and carbon sequestration by vegetation and forest growth and is measured in million t CO₂-eq·yr⁻¹. Production includes the generated output from agriculture and timber production and is measured in billion £·yr⁻¹. Biodiversity is quantified with a unitless indicator that combines species occurrence probabilities and habitat condition values and is normalised between 0 and 1 in each cell.”

And the Heading of Table 1 in Line 953 (Line 971 in the tracked changes version):

“Performance if each of the objectives was minimized and maximised, with no limits on the area of land-use changes”

Reviewer 2 – Comment 4: Line 154: why 35-35-30 instead of 33-33-33? Or will it be large difference between these two? Additionally, your normalization seems 0-1 in Fig. 1, why weighting combinations here is 0-100? Are these numbers percentages?

Response:

Thanks for asking for clarification on this point. We recognise how additional explanations could make this a lot clearer. The selection of 35-35-30 instead of 33-33-33 is caused by the chosen step size of 5 for the weightings instead of a step size of 1. This step size was selected to balance the number of scenarios and therefore their comprehensibility, computational efficiency, and resolution. A finer step size (e.g., one or two) would have significantly increased the number of scenarios generated

(step size 10 -> 66 scenarios;

step size 5 -> 231 scenarios;

step size 2 -> 1326 scenarios;

step size 1 -> 5151 scenarios) and the computational complexity without substantial improvements in the results, while a larger step size (e.g., ten) would have led to excessive discretization, which does not allow the creation of a smooth pareto frontier. The overall land allocation conversion outcomes also do not change significantly when a smaller step size is chosen. We have added an explanation in the Results section in Lines 166-171 (Lines 169 – 173 in the tracked changes version) and the Methodology section in Lines 901 - 907 to make that clearer, as well as the pareto frontier resulting from a step size of 10 and 2 in the SI in Figures S1 and S2 and the resulting conversion hotspots maps for step size of 10 and 2 in Figures S3 and S4. Regarding the normalisation scale, there is only an indirect connection between the normalised results and the weighting combinations. The weighting combinations (e.g., 35-35-30) are expressed as percentages (i.e., summing to 100) to represent the relative importance of each objective in the weighted sum optimisation, while the normalised results represent the resulting overall benefit normalised against the lowest possible and highest possible outcomes. We have clarified this in the manuscript in Lines 175-182 (Lines 177-184).

Revised text in Lines 166-171 (Lines 169 – 173 in the tracked changes version):

“We discretise the continuous weighting combinations into a step size of five, where 100%-0%-0% would fully prioritise the first objective while 35%-35%-30% would consider all three objectives nearly equally, resulting in 231 vectors of 3-way weightings for each cell. The step size was chosen based on some preliminary tests to balance the number of scenarios and the resolution needed to create a smooth pareto frontier (see Figure S1 – Figure S4).”

And in lines 175-182 (Lines 177-184):

“Additionally, it is worth noting that there is only an indirect connection between the normalised benefit results (normalised between 0 and 1) and the objective weightings (expressed as percentages). While an objective weighting of 100%-0%-0% leads to the maximum outcome for the first objective, it does not necessarily result in a minimum for the other two objectives, as there may be synergies between them. For example, a weighting of 100% on carbon sequestration will result in the normalised maximum value of 1 for carbon sequestration outcomes, while still yielding considerable benefit outcomes for biodiversity due to the synergies between the two objectives.”

And in Lines 883-889 (Lines 901 – 907):

“The step size of five was chosen as a trade-off between resolution and the comprehensibility of the number of scenarios, as well as the computational runtime. A smaller step size would have increased the number of scenarios significantly, which makes exploring them infeasible and increases the computational runtime, while a larger step size would have led to excessive discretisation and made it impossible to draw the full pareto-frontier (see Figure S1 – Figure S4). Preliminary tests showed that a step size of five was able to create a smooth pareto frontier while keeping the number of scenarios manageable.”

Reviewer 2 – Comment 5: Line 190-213: I was initially wondering what the changes in the performance values means in real conservation efforts, until I kind of found the answer in the next paragraph (line 214). It would be better to make it clear what changed in the performance represent for each objectives, e.g., change in 0.1 of performance in production represent xx billion £·yr⁻¹, or change in xx million t CO₂-eq·yr⁻¹ carbon sequestration, if my understanding is correct. Additionally, the author may also need to explain the chosen of unit in each objective. I realized the details could be in the method section, while given the current structure of the paper, I prefer it be readable without checking the method since method is in the end of the paper.

Response:

Thank you for your comment. We agree that it's important to clarify what the changes in scenario performance mean in real numbers.

To address this, we have included the numbers in actual units next to the normalised values in Lines 230-245 (Lines 232-252 in tracked changes version). Additionally, we simplified the paragraph by moving some of the normalised numbers into a Table in the SI (Table S3)

We also included an explanation of the chosen units at the beginning of the results section in Lines 155 - 162 (which also connects to your 3rd comment) to make the results section clearer on its own.

Revised Text in Lines 155-162 (Lines 157-164 in tracked changes version):

“For each grid cell, we identified the land-use conversions that maximise and minimise each of the three objectives: Carbon sequestration, (food and timber) production and biodiversity. The performance of a land allocation scenario in terms of carbon sequestration consists of carbon fluxes from land conversion, emissions from agriculture and carbon sequestration by vegetation and forest growth and is measured in is measured in million t CO₂-eq·yr⁻¹. Production includes the generated output from agriculture and timber production and is measured in billion £·yr⁻¹. Biodiversity is quantified with a unit less indicator that combined species occurrence probabilities and habitat condition values and is normalised between 0 and 1 in each cell.”

And in Lines 230-245 (Lines 232-252 in the tracked changes version):

“The point where the pareto frontier is met when increasing production without changing the other two objectives has a production **value 23.6% higher (+3.74 billion £·yr⁻¹)** than the current (see Figures 1a & b) without decreasing biodiversity or carbon sequestration. For the other two objectives, there is no intersection with the pareto frontier when increasing one objective while keeping the other two constant. Instead, when increasing carbon sequestration while keeping production constant, biodiversity will increase as well. When increasing carbon sequestration as much as possible without decreasing production, **the intersection point with the frontier shows an improvement of +128.9% (an additional 71.9 million t CO₂-eq·yr⁻¹)** compared to the current, meaning shifting from significant carbon emissions to a relatively low level of carbon sequestration, and comes with an improvement of **+22,530 biodiversity indicator points (see Error! Reference source not found.a & c)**. When increasing biodiversity as much as possible without decreasing production, the intersection point with the frontier shows an increase of **14.2% (+31,920 biodiversity indicator points)** compared to the current biodiversity performance, which comes with an improvement of carbon sequestration of **37.9 million t CO₂-eq·yr⁻¹** (see Error! Reference source not found.b & c). The distances between the current performance and the points on the pareto frontier are visualised in Error! Reference source not found., and all normalized and actual numbers can be found in **Table S3.**”

And the new Table S3 in the SI:

Table S3 Distance from the current land use performance to the pareto frontier

		Current	Distance to pareto frontier	Percentage Improvement	Synergies	
Carbon	Normalised	0.36	0.30	+128.9%	with Biodiversity	0.12
	Mio. t CO ₂ -eq·yr ⁻¹	-55.78	71.9			+22,530 Biodiv. Indic. points
Production	Normalised	0.62	0.15	23.6%	None	-
	Billion £·yr ⁻¹	15.8	3.74			-
Biodiversity	Normalised	0.34	0.17	14.2%	with Carbon seque.	0.16
	Biodiv. Indic. points	225,000	31,920			37.9 mio. t CO ₂ -eq·yr ⁻¹

Table S3 Distance from the current land performance to the pareto frontier along each axis as a measure of the inefficiency of current land use

Reviewer 2 – Comment 6: Line 223: would be good to explain what “biodiversity score” means or represent

Response:

We recognize that the term “biodiversity score” needs to be explicitly defined at the beginning of the results section to ensure clarity without the need to go to the Methodology section. The biodiversity score we use is a combination of species occurrence probabilities from a species distribution model and habitat condition values. We have added an explanation at the beginning of the results section in line with our edits made for your third comment.

Revised text in Lines 160-162 (Lines 162 – 164 in the tracked changes version):

“Biodiversity is quantified with a unitless indicator that combines species occurrence probabilities and habitat condition values and is normalised between 0 and 1 in each cell.”

Reviewer 2 – Comment 7: Line 285: provide more details on the setting of budget gradient, how it was defined and measured, why it is in percentage, what it means in real management, 1% vs. 100%?

Response:

Thank you for your comment. We recognise that the concept of the budgets requires further clarification to enable readers to understand our results section as a stand-alone section and have added a more detailed explanation at the beginning of the conversion budget results section, in Lines 321- 332 (Lines 328 – 339 in tracked changes version).

The budget represents the share of rural land that is made available for land-use conversions. So while 1% represents a relatively conservative land conversion scenario, higher budgets represent more ambitious amounts of conversions, and the highest shares up to 100% show a more theoretical exploration to show what the most extreme case with the option to convert all rural land would look like. It is in percentage because it represents the share of rural land. The revised text now includes a more detailed clarification.

Revised text in Lines 321- 332 (Lines 328 – 339 in tracked changes version):

“This analysis was carried out by limiting land-use change to a budget of change, meaning only a certain share of the land classified as convertible can be converted. For each combination of weights, we identify the ranked list of conversions across the entire country, based on the weighted sum of their performance. We then identify how many of those ranked conversions can be implemented within the total ‘budget’ of permitted conversions. **Cells with the most prominent overall benefits under a weighting combination are chosen first, while less beneficial cells are added in merit order with higher budgets.** The analysis was done for a range of conversion budgets, representing the share of convertible land that is made available for conversion. This enables decision makers to explore how much change would be necessary to achieve certain outcomes and therefore how ambitiously land conversion targets need to be set. To explore a broad range of conversion rates, 11 budgets were analysed, starting with a budget of 1% and then stepping in increments of 10%, up to 100% of the rural land available for conversion.”

Reviewer 2 – Comment 8: Line 300: consider adjusting the figure and the color gradient. I can not tell 10% from 20%, and whether the shape is linear or convex.

Response:

This is very useful feedback regarding the colour gradient of the figure. To address this, we have made several changes:

1. Since even a 100% budget only leads to maximum values of 72% of rural land changed (more would not actually lead to further improvements), we have removed the labels for 80 and 90 % which leads to more spread-out colours for the remaining lines.
2. We have added labels showing the budget next to the lines to make it easier to identify each line.

We have updated the plot in Line 336 (Figure 3):

Reviewer 2 – Comment 9: Line 311: are there any turning points of budget for that?

Response: This is indeed a very interesting question. For the synergies between production and biodiversity that you are referring to, we couldn't identify any turning points throughout the budgets. As for the other two subplots, additional changes can only be created up to conversion rates of about 70%, any further change would not lead to additional benefits or even be counter-productive. We have added these insights to the section. Additionally, we think the clarifying changes we made to the figure will help the reader interpret the plot accordingly.

The revised text in Lines 362-363 (Lines 370 – 371 in the tracked changes version):

“This can be explained by the lack of synergies between the two objectives - the best choice for production will not be beneficial for biodiversity and the other way around. Compared to Figures 3a & c, the frontiers throughout the range of budgets are much closer together, meaning that an increase in budget is not as beneficial for managing the trade-offs between production and species occurrence as for the other two combinations. ***This pattern can be seen throughout all the budgets without any clear turning points.***”

And the adjusted plot (see comment above) in Line 336.

Reviewer 2 – Comment 10: Line 695: what are the 88 species and are they representative enough?

Response:

This is indeed an important question that needs further explanation. We acknowledge the need to clarify which types of species were included in the analysis and how well they represent biodiversity patterns. The species were chosen based on the list used in another big land-use model called NEVO, which based its selection on several criteria such as the number of records in the last 20 years, the spreading of the species or its independence from water quality. We have revised the Methodology section to include this information. Additionally, we've made the following revisions:

- We have provided a summary of the number of species in each taxonomic group in the main manuscript

- We have added a full list of the species in the supplementary information, in Table S6

Revised text in Lines 829-835 (Lines 845 – 851 in the tracked changes version):

*“The species distribution model was **developed** in line with Croft et al. (2017)⁹² using **86 species from the BAP priority species list**⁹³, including **38 vascular plants, 13 mammals, 22 invertebrates, five lichens, one herptile and seven birds**. A detailed list of all species can be found in Table S6 in the SI. This selection was based on a list of 100 priority species that have also been used in the SDM run as part of the NEVO land-use model⁴² and was composed based on a range of criteria, including the number of records in the last 20 years, the spreading of a species and independence from water quality⁹⁴.”*

And the full list of species in Table S6 in the SI.

Reviewer 2 – Comment 11: Line 709: how was the calculations are actually done? The reader may need to understand this without checking through another published paper.

Response:

Thank you for pointing this out. We acknowledge that it is not entirely clear here what we mean and that referring to another published paper without sufficient explanation may make it difficult for readers to fully understand our methodology. What is meant is that the species occurrence probabilities within each cell are summed up over all species. To make this clearer, we have edited the manuscript by adding the equation describing the procedure in Lines 847-849 (Lines 864 – 865 in the tracked changes version).

Revised text in Lines 847-849 (Lines 864 – 865 in the tracked changes version):

*“The resulting occurrence probabilities for all species were summed up to an overall indicator ($\sum_{species} P(cell, species)$, where **P is the occurrence probability**), as suggested by Calabrese et al. (2014).”*

Reviewer 3:

This manuscript addresses a critical topic concerning land-use optimization for Great Britain, providing valuable insights into balancing carbon sequestration, agricultural productivity, and biodiversity conservation. The authors utilize robust multi-objective optimization methodologies, present novel high-resolution analyses, and demonstrate clear implications for national policy-making. However, substantial revisions are needed to enhance readability, better justify methodological choices, and strengthen the discussion on policy implications and limitations.

The manuscript, while methodologically sound, suffers from overly complex descriptions in parts, lacks sufficient justification for some key assumptions, and would benefit significantly from more refined visualizations and clearer interpretation of the results. A revised version must address these clarity and justification issues explicitly.

Reviewer 3 – Comment 1: Lines 18-31: Clearly state your novel contribution to distinguish your research from other existing work explicitly in the abstract.

Response:

Thank you for this helpful suggestion. We have now revised the abstract to more explicitly highlight the key novel aspects of our work- specifically, the integration of multiple land-based objectives into a spatial optimisation framework that explores the full option space instead of a few scenarios.

The revised text in Lines 26 - 30:

*“Here, we evaluate the trade-offs between three objectives for rural land: agricultural/forestry production, carbon sequestration and biodiversity, by calculating metrics for these three objectives on a 500mx500m grid covering Great Britain (GB). We use a multi-objective optimisation **that allows us to explore the full option space of possible land conversions and identify the land allocations that satisfy a broad range of priority weightings between the three objectives and therefore entail limited trade-offs.** Our results show that the current land-use in GB is far from optimal for any combination of objectives. **We identify the locations where it is possible to significantly improve carbon sequestration and biodiversity, even with a relatively small proportion of the land being converted to other uses, without compromising overall agricultural production, provided conversions are located carefully.**”*

Reviewer 3 – Comment 2: Lines 69-82: Clarify explicitly why leaving the EU's Common Agricultural Policy provides a unique and timely opportunity for your research, linking this context more strongly to your objectives.

Response:

We have revised the Introduction to clarify how the UK's departure from the EU's Common Agricultural Policy represents a significant policy shift that creates a rare and timely opportunity to rethink land-use strategies. This policy transition allows the UK to redefine its agricultural subsidies and environmental priorities, and new policies and incentives are being designed. This aligns with our study's objective of exploring how land-use can be optimised to deliver multiple objectives (carbon sequestration for climate mitigation, food production and biodiversity). We now emphasize this in Lines 82-84 and 144-146 (Lines 146-148 with tracked changes).

Revised text in Lines 82-84:

*“...environmental land management instruments to achieve environmental objectives like climate mitigation and adaptation³². **Making sure that new policies and land-use strategies contribute to achieving climate and biodiversity objectives without threatening food production is of high interest.** At the same time, the evidence needed to guide new land-use policies is missing³³.”*

And in Lines 144-146 (Lines 146-148 with tracked changes):

*“By analysing the full range of possible land-use conversions in the country, we enable decision-makers to explore the entire options space **as a basis for designing new targeted land-use policies.**”*

Reviewer 3 – Comment 3: Lines 111-122: More clearly articulate the gap in existing modelling literature that your spatially explicit analysis addresses.

Response:

Thanks for this suggestion. The gap our paper addresses includes two aspects: 1) a lack of papers that address the triple challenge of carbon sequestration, food production and biodiversity with limited land for national policy making and 2) a lack of a spatially explicit model that shows trade-offs by exploring an extensive space of land conversion options targeted for policy making on a national scale. Most existing models are either non-spatial calculations that can't show spatial trade-offs and patterns; spatial studies for a small region; they combine all objectives into one measure (for example, by expressing all ecosystem services monetarily) and/or only compare two or three alternative scenarios. Our study adds to the literature by taking a different approach to exploring potential land-use conversions for multiple objectives. We are addressing

the gap in the literature by examining a broad range of potential land-use conversions based on 231 different weighting combinations of our objectives, and for a range of potential conversion budgets. *Instead of designing a restricted set of scenarios, we explore the entire range and construct the trade-off surfaces. This is rarely done and is targeted at informing the formulation of policy targets along the respective trade-offs.* By analysing this large space of priority-based scenarios, we can identify the pareto-frontier of most beneficial conversions depending on the priority weightings. Analysing the spatial patterns that a broad range of scenarios have in common allows us to identify conversions in certain regions that show relatively low trade-offs and are robust to changing weighting combinations. To address your comment, we have revised the Introduction to make the gap in the literature that we are addressing clearer.

The revised text in Lines 129-130 (Lines 131 – 133 in the tracked changes version):

“There is a lack of existing studies that address the triple challenge of carbon sequestration, food (and timber) production, and biodiversity on limited land for national policy making. Additionally, existing models do not provide a spatial overview of trade-offs for national policymaking that includes the full range of land-use possibilities without limiting the outputs by only exploring a handful of scenarios or influencing the outcomes with predefined weightings. Therefore, we present a spatially explicit approach targeted at decision-makers in GB that shows the entire decision space for the full range of potential priorities while pointing out synergies and trade-offs that need to be considered.”

Reviewer 3 – Comment 4: Lines 152-154: Explain explicitly why you chose the step size of five for the weighting discretization and justify why alternative step sizes were not considered.

Response:

This is a good point. We have now added a more detailed explanation for the choice of a step size of five in the weighting discretization. This step size was selected to balance the number of scenarios and therefore their comprehensibility, computational efficiency, and resolution. A finer step size (e.g., one or two) would have significantly increased the number of scenarios generated (step size 10 -> 66 scenarios;

Step size 5 -> 231 scenarios;

step size 2 -> 1326 scenarios;

step size 1 -> 5151 scenarios) and the computational complexity without substantial improvements in the results, while a larger step size (e.g., ten) would have led to excessive discretization, which does not allow the creation of a smooth pareto frontier.

To clarify our choice, we have updated the manuscript in the Results section (Lines 169-171 (Lines 171-173 in the tracked changes version):) and the Methods section (Lines 883-889 and Lines 901-907 in the tracked changes version). Additionally, we have added the resulting plot of the frontiers and the land allocation for a step size of 10 and 2 in the Appendix, to show what the effect would be.

Revised Text in the Methodology in Lines 883-889 (Lines 901-907 in the tracked changes version):

“The step size of five was chosen as a trade-off between resolution and the comprehensibility of the number of scenarios as well as the computational runtime. A smaller step size would have increased the number of scenarios significantly, which makes exploring them infeasible and increases the computational runtime, while a larger step size would have led to oversimplification and made it impossible to draw the full pareto-frontier (see Figure S1 – Figure S4). Preliminary tests showed that a step size of five was able to create a smooth pareto frontier while keeping the number of scenarios manageable.”

And in the results Section in Lines 169-171 (Lines 171-173 in the tracked changes version):

“The step size was chosen based on some preliminary tests to balance the number of scenarios and the resolution needed to create a smooth pareto frontier (see Figure S1 – Figure S4).”

Reviewer 3 – Comment 5: Lines 186-207: The detailed numeric descriptions (percentages and values) in-text are overly complex. Consider summarizing these more concisely, shifting detailed numeric descriptions into supplementary materials or tables.

Response:

Thanks for your suggestion. We agree that the level of detail in the numeric descriptions in this Section could be reduced to make it more readable and focus on the key information. Therefore, we have now edited the section to only show the percentage improvement and improvement in actual units and moved the rest of the information into Table S3 in the SI.

This is the revised text in Lines 228 – 245 (Lines 230-252 in the tracked changes version):

*“To identify the inefficiency of the current land-use in relation to the pareto frontier, the distance between the normalised current performance and the frontier was measured parallel to each of the three axes. The point where the pareto frontier is met when increasing production without changing the other two objectives has a production **value 23.6% higher (+3.74 billion £·yr⁻¹)** than the current (see Figures 1a & b) without decreasing biodiversity or carbon sequestration. For the other two objectives, there is no intersection with the pareto frontier when increasing one objective while keeping the other two constant. Instead, when increasing carbon sequestration while keeping production constant, biodiversity will increase as well. When increasing carbon sequestration as much as possible without decreasing production, the intersection point with the frontier shows an improvement of **+128.9% (an additional 71.9 million t CO₂-eq·yr⁻¹)** compared to the current, meaning shifting from significant carbon emissions to a relatively low level of carbon sequestration, and comes with an improvement of **+22,530 biodiversity indicator points** (see **Error! Reference source not found.a** & c).. When increasing biodiversity as much as possible without decreasing production, the intersection point with the frontier shows an increase of **14.2% (+31,920 biodiversity indicator points)** compared to the current biodiversity performance, which comes with an improvement of carbon sequestration of **37.9 million t CO₂-eq·yr⁻¹** (see **Error! Reference source not found.b** & c). The distances between the current performance and the points on the pareto frontier are visualised in **Error! Reference source not found.**, and all normalized and actual numbers can be found in **Table S3**”*

And the new Table S3:

Table S3 Distance from the current land use performance to the pareto frontier

		Current	Distance to pareto frontier	Percentage Improvement	Synergies	
Carbon	Normalised	0.36	0.30		with	0.12
	Mio. t CO ₂ -equ.yr ⁻¹	-55.76	71.9	+128.9%	Biodiversity	+22,530 Biodiv. Indic. points
Production	Normalised	0.62	0.15	23.6%	None	-
	Billion £·yr ⁻¹	15.8	3.74			
Biodiversity	Normalised	0.34	0.17		with	0.16
	Biodiv. indic. points	225,000	31,920	14.2%	Carbon sequ.	37.9 mio.t CO ₂ -equ.yr ⁻¹

Table S3 Distance from the current land performance to the pareto frontier along each axis as a measure of the inefficiency of current land use

Reviewer 3 – Comment 6: Lines 226-264: Explain explicitly why converting semi-natural habitats to pasture is beneficial in your model since such conversions typically imply biodiversity losses. Clarify this seemingly counterintuitive result.

Response:

Thank you for pointing this out. We agree that, ecologically, converting semi-natural habitats to pasture might lead to decreased biodiversity and must be seen critically. However, in our model, these conversions are selected under scenarios where production is weighted more heavily. The section you are commenting on is looking at only strictly better scenarios – so scenarios where all three objectives have to be better or the same as currently. This includes production, which is already prioritised relatively highly in the current land-use, so further improvements of production lead to those extreme conversions. Therefore, this result reflects the trade-offs of the mathematical optimisation under the assigned weights that lead to strictly better scenarios and is not necessarily a recommendation. The effect might be increased because we have not included rough grazing on semi-natural habitats. Therefore, the production score of semi-natural grassland might be lower in our model than in reality.

We are discussing the effect of not including rough grazing in Lines 524-529 in the discussion and have now added to the manuscript in Lines 276-289 (Lines 283 – 296 in the tracked changes version) in the Results Section to emphasise that these outcomes reflect the trade-offs under more production focused priority weightings.

The revised text in Lines 276-289 (Lines 283 – 296 in the tracked changes version):

*“...central Wales (Gwynedd and west Powys) and parts of Scotland (in the Highlands, northern Perthshire, western Aberdeenshire, and northern Angus). **While conversions from semi-natural grassland to pasture are counterintuitive from an ecological perspective, they reflect the trade-offs under more production-focused priority weightings. One reason for this is the relatively low biodiversity benefit from semi-natural grasslands in our model. Additionally, most of these areas are most likely already used for extensive grazing on acid grassland, heather and heather grassland. Therefore, maintaining and expanding livestock grazing in these areas would come with relatively small disadvantages for biodiversity and carbon targets compared to the current pasture in the south of England, which could offer valuable benefits as arable land or natural habitat. It is important to remember that the subset of strictly better scenarios does not allow a decrease in agricultural production and, therefore, also includes conversions to arable land and pasture where sensible. Looking at the clear current prioritisation of agriculture over nature and the ambitious environmental objectives, the strictly better scenarios that do not allow any decrease in overall production might be considered unambitious and slightly more biodiversity and carbon-focused changes might be considered appropriate.**”*

Reviewer 3 – Comment 7: Lines 437-465: Provide a more explicit and structured discussion on practical challenges (e.g., political feasibility, socio-economic impacts, cultural acceptance) associated with the suggested land-use transitions.

Response:

Thanks for this suggestion. We have added to our discussion of practical challenges by talking about the specific challenges of the conversions to each of the land-use categories. Additionally, we have added more challenges, such as land ownership and the costs of land conversions, to our Discussion in Lines 584-609 (Lines 595 – 620 in the tracked changes version).

This is the revised text in Lines 584-609 (Lines 595 – 620 in the tracked changes version):

*“While the locations and changes we identified are the most beneficial from a biophysical perspective, local needs such as jobs or access to nature and cultural preferences must be considered in subsequent, more detailed studies aiming to design concrete policy implementation. **The conversion of large areas of pasture in an area might cause some resistance due to jobs in farming being lost and the characteristic look of landscapes changing. This might not only be challenging for financial reasons but also interfere with the identity and culture in certain areas that are characterised by farming. At the same time large scale conversions of grassland to arable land in other regions might affect access to nature and the aesthetic of the landscape and might therefore face resistance. Additionally, other factors might affect the feasibility of implementing the results of our analysis, such as land ownership or local political context and capacities. The economic costs of land conversion are another constraint on implementation that has not been included in this analysis. While the agricultural production calculated for this study gives an idea about where opportunity costs for farmers are higher in the case of conversion to other area, it does not include the actual cost of new policy schemes for incentivising landowners, the cost of conversion to different land-use types or other cost that might occur in the implementation. Higher conversion rates might be less feasible when these constraints are included. Additionally, time delays from the time it takes for land-uses and ecosystems to establish and deliver their full benefits have not been considered in this study and might affect the real-life outcomes from land conversions. Future work could include these aspects to investigate which conversions are not only beneficial but also feasible when including factors such as land ownership and economic costs of land conversion.**”*

Reviewer 3 – Comment 8: Lines 466-480: Discuss how different assumptions around carbon sequestration in timber products might alter policy recommendations, given their significant role in your findings.

Response:

We agree that assumptions around carbon sequestration in timber products influence the potential benefits from plantation forests. However, in our analysis, plantation forests don't come up that often which is to some degree caused by our assumptions of a relatively small amount of carbon stored in harvested wood products. The second reason is that they offer relatively low biodiversity value in our model and are therefore deprioritized, especially under objective weightings that include biodiversity.

Given their relatively small benefits in terms of biodiversity, even higher carbon storage assumptions would only have a limited effect. Nevertheless, we have added to the Discussion to clarify this.

Revised text in Lines 533-534 (Lines 544 – 545 in tracked changes version):

*Changes in the life cycle and usage of timber could change that and make plantations more valuable for long-term carbon sequestration. **However, due to the much higher biodiversity benefit, natural forests will remain the superior option because of the strong synergies they offer for biodiversity.**”*

Reviewer 3 – Comment 9: Lines 481-506: Expand your discussion on the potential impacts on food security, particularly addressing the trade-offs involved if dietary changes are not realized.

Response:

To address this, we have now expanded our discussion on trade-offs with food security if dietary changes are not realised.

Revised text in Lines 548-571 (Lines 559 – 582 in the version with tracked changes):

“While our model explores the interaction between ecological objectives and food production, it does not assume or enforce any specific dietary shifts. If current diets persist, especially in terms of their high consumption of animal products, which are very land-intensive, the conversion of agricultural land to other land-uses comes with severe trade-offs. This is in line with Bajželj et al. (2014)⁵⁵ and Springmann et al. (2018)⁵⁶ who show that achieving climate and biodiversity targets without dietary shifts will be challenging. A reduction of land for food production, which occurs in many scenarios other than the most production-oriented, including the production of animal products, without a change in diets, can, to some degree, be compensated by closing the gap in efficiency. However, after this gap is closed, any further reduction will lead to an increase in imports, potentially from countries less committed to environmental targets or richer biodiversity and, therefore, outsourcing of emissions and biodiversity degradation. Most land-use studies base their more sustainable scenarios on significant dietary changes^{34,57}, which is a desirable objective and an essential step towards sustainability but still uncertain. The National Food Strategy aims to tackle the “environmental damage caused by intensive agriculture”⁵⁸, but until then, an approach like the one presented in this paper can help focus land conversion on areas where trade-offs and losses in food production are minimised. Therefore, we highlight the importance of integrated policy approaches that combine land-use optimisation with a transformation of the food system, sustainable consumption and minimise food waste to further decrease trade-offs and ensure sustainable outcomes. Additionally, it needs to be pointed out how the ratio between arable land and pasture, and therefore the share of animal products in the overall production, differs throughout the scenarios. The current scenario has a ratio of 1.2 between land for arable production and pasture, while the pareto-efficient scenarios show ratios between 0.11 and 3.0, with a mean value of 1.11.”

Reviewer 3 – Comment 10: Lines 507-522: Clearly discuss the potential limitations arising from using the selected biodiversity indicators. Highlight how including habitat connectivity might change your recommendations.

Response:

We acknowledge that our biodiversity indicators, which are based on species occurrence and habitat condition (in common with many other indicators), cannot represent all aspects of biodiversity, such as connectivity. While it wasn't possible computationally to include connectivity in the optimisation, it might lead to different spatial configurations favouring larger patches or forming corridors between areas with high biodiversity scores. We have expanded the Discussion of this limitation in Lines 621- 625 (Lines 634 – 638 in the version with tracked changes) to better emphasise that specific ecological processes that rely on connectivity, such as movement patterns, gene flow, or metapopulation dynamics, are not captured by our biodiversity indicator. Additionally, we have added that including a connectivity measure might prioritise habitat corridors and larger habitat patches over isolated biodiversity-rich cells.

We are investigating the implications of habitat connectivity on biodiversity in a separate, smaller-scale study that we are currently working on.

The revised text in Lines 621- 625 (Lines 634 – 638 in the version with tracked changes):

“Especially for biodiversity, using different indicators can lead to differing results. For computational reasons, it was impossible to include habitat connectivity in the biodiversity indicator, which might have led to **a shift of the spatial patterns created, prioritising larger habitat patches and habitat corridors over isolated cells.**”

Reviewer 3 – Comment 11: Lines 523-539: Clarify explicitly the implications of not considering urban expansion in your model and how future urban development pressures might affect the feasibility of your land-use optimization.

Response:

Thank you for raising this important consideration. We would like to clarify that this issue is already acknowledged in the Discussion section of the manuscript in Lines 640-648 (Lines 654 – 662 in version with tracked changes). We note that our model does not incorporate future urban expansion, which is likely to happen on the green belt areas around existing urban areas. However, we have extended this section to clarify that this affects the real-world feasibility of our results in these areas.

The revised text in Lines 640-648 (Lines 654 – 662 in version with tracked changes):

*“We have assumed urban land-use to remain unchanged, and it has not been evaluated for any of its benefits. However, urban expansion and housing development are predicted to take up additional land in the UK⁵⁹, and some are arguing that this should happen, especially on the Green Belts around existing urban areas⁶⁰. That will require the conversion of current rural areas and natural land, **which puts additional pressure on the land and affects the feasibility of our model outputs in these areas. To minimise this uncertainty and an additional source of trade-offs, it has been recommended to keep new urban development compact³⁴. These effects are currently not part of the model but can be included with urban expansion scenarios in a future version.**”*

Reviewer 3 – Comment 12: Lines 580-614: Clarify the rationale behind not excluding protected areas explicitly from your potential land conversions. Discuss potential implications or biases that might result from this decision.

Response:

Thanks for your comment, we appreciate the opportunity to clarify this point. In our analysis, we did not explicitly exclude protected areas from the potentially convertible land for two main reasons:

1. Our primary objective was to explore the theoretical spatial trade-offs and synergies across our three objectives without imposing legal or institutional constraints.
2. Protected area boundaries often include arable land or pasture, so they are relevant for our analysis

We acknowledge that including protected areas as convertible may lead to overestimating the potential for land conversion in regions that are unlikely to be altered drastically in practice. We have now added to the discussion section, noting this limitation and suggesting that future applications of the framework could constrain the optimisation by including legal or policy-based restrictions.

Revised text in the Methodology in Lines 728-730 (Lines 744 – 746 in version with tracked changes):

*“An exclusion of land for conversion based on land classifications such as national parks, battlefields, etc., was not considered to explore the full **theoretical space of possibilities without any current institutional constraints.**”*

And in the Discussion in Lines 648-653 (Lines 662 – 668 in version with tracked changes):

“Furthermore, we did not exclude protected areas and other classified areas from consideration for land conversions to explore the full theoretical space of options without imposing current institutional constraints. Protected areas, for example, can include arable land or pasture and are relevant for our trade-off analysis. However, we acknowledge that this might lead to an overestimation of the feasibility of land conversions in certain areas. Future work could include these institutional and legal limitations.”

Reviewer 3 – Comment 13: Lines 615-657: Provide additional details on the assumptions underpinning your carbon sequestration calculations, explicitly acknowledging the uncertainties related to emissions from pastures and plantations.

Response:

Thanks for raising this point. We agree that we could add more detail about the exact assumptions used for the carbon sequestration model for pasture and forests. We added more details about the assumptions we made in the Methodology section and added Table S4 with the exact numbers used for our livestock calculations and Table S5 with the exact assumptions for the different forest types in the SI.

While our model includes emissions from livestock (e.g., methane emissions) and soil carbon loss from land conversion, we recognise that several factors introduce uncertainties, such as varying grazing densities and management practices. We have revised the Methodology section to explicitly state these uncertainties.

The revised text in Lines 751-763 (Lines 769 – 779 in version with tracked changes):

*“For arable land emissions from fertilisers, an average of 1460.67 g N₂O-N·ha⁻¹ was assumed based on the emissions reported by Bell et al. (2015)⁶⁵, which converts to 626.63 kg CO₂ eq·ha⁻¹. **Differences between management practices or soil composition have not been included.***

To calculate the emissions from pasture, the average livestock density per meadow and pasture area between 2016 and 2020 was taken from FAOSTAT⁶⁶ and converted using the livestock unit coefficients used by Eurostat⁶⁷. Shares for beef and dairy cows from the UK gov livestock statistics⁶⁸ were applied. Emission values for beef cows, dairy cows, and sheep for CH₄ and N₂O⁶⁹ were averaged and converted to CO₂-eq using GWP-100 conversion values⁷⁰. Emissions from livestock per ha on pasture sum up to 7.62 t CO₂ eq·ha⁻¹yr⁻¹, assuming an average livestock mix on all pasture areas. **All detailed numbers can be found in Table S4. In reality, beef cows, dairy cows and sheep would not be grazing on the same land patch and stocking densities and management practices might differ substantially between farms and regions.**”

And Lines 770-792 (Lines 786 – 808 in version with tracked changes):

*“To calculate the sequestration and emissions from forest plantations and natural broadleaved forests, tree species occurrence maps from the European Forest Institute^{76,77} **were obtained for six coniferous and seven broadleaved tree species.** These were combined with location-specific yield maps from the Forest Research Ecological Site Classification⁷⁸ and the **cumulative total sequestered carbon values** per yield class from the woodland carbon code⁷⁹. **No***

thinning is assumed for broadleaved forests, while thinning is assumed in the management of coniferous plantations. Biomass that has been removed due to thinning is added to the cumulative sequestration. For managed plantation forests, felling times were assumed based on the tree and yield class specific age of maximum mean annual volume increment⁸⁰. The time scale of carbon sequestration and the resulting differences between already established forests and new conversions to forests are included by considering the tree age distribution⁸¹, the species and yield specific growth rates and for plantation forests the resulting time until felling. **The age distribution of newly planted forests is assumed to result from consistent planting between the baseline year 2020 and 2050. All assumptions for newly planted and existing broadleaved forests as well as coniferous plantations are summarised in Table S5.** To estimate the ratio of long-term carbon storage in harvested wood products, the global wood product model by Peng et al. (2023)⁵² was recreated for the UK using the FAOSTAT Forestry Production and Trade data for the UK in 2022⁸² and the Forest Research Forestry Statistics 2023^{83,84}. Average yearly numbers for carbon sequestration/emissions from plantation woodlands and natural broadleaved forests calculated over the period 2020-2050. **For plantation woodlands, this number is then multiplied by the share of long-term carbon storage in harvested wood products. Our assumptions are subject to a number of uncertainties regarding differences in management practices, harvesting patterns, the exact tree species composition or the choice of tree species being planted in afforestation projects.”**

Reviewer 3 – Comment 14: Lines 658-683: Clarify explicitly why and how you chose the species used for the biodiversity modeling. Discuss any limitations or potential biases inherent in these selections.

Response:

Thank you for this question. We acknowledge the need to clarify which types of species were used for the biodiversity modelling and what their limitations or biases might be. The species were chosen based on the list used in another big land-use model called NEVO, which based their selection on several criteria such as the number of records in the last 20 years, the spreading of the species or its independence from water quality. We have revised the Methodology section to include this information. Additionally, we've made the following revisions:

- We have provided a summary of the number of species in each taxonomic group in the main manuscript
- We have added a full list of the species in the supplementary information, in Table S6

Revised text in Lines 829-835 (Lines 845 – 851 in the version with tracked changes):

“The species distribution model was developed in line with Croft et al. (2017)⁹² using 86 species from the BAP priority species list⁹³, including 38 vascular plants, 13 mammals, 22 invertebrates, five lichens, one herptile and seven birds. A detailed list of all species can be found in Table S6 in the SI. This selection was based on a list of 100 priority species that have also been used in the SDM run as part of the NEVO land-use model⁴² and was composed based on a range of criteria, including the number of records in the last 20 years, the spreading of a species and independence from water quality⁹⁴.”

And the full list of species in Table S6 in the SI.

Reviewer 3 – Comment 15: Lines 684-724: Clearly outline the multi-objective optimization procedure step-by-step, providing a simplified illustrative example that helps readers unfamiliar with such methods better understand your analytical process.

Response:

Thank you for your feedback. We want to clarify that the optimisation is explained in Lines 868-910 and visualised in Figure 6. However, we have edited the explanations to make the procedure easier for readers unfamiliar with such methods. Additionally, we have added numbers to the steps in the plot and revised the plot description to explain each of the numbered step in more detail

Revised text in Lines 890-907 (Lines 908 – 918 in the version with tracked changes):

“The steps of the optimization are visualized in Figure 1. After calculating the location specific potential benefits of conversion to each of the four land-use categories (as described above), the benefit values are normalized for each benefit across all four land-use categories.

Subsequently, for each constellation of weights, the following steps were implemented to create a scenario maximizing the overall weighted benefit:

With w_O as the weight of each objective O in the scenario, $O = C, P, B$ (carbon, production, biodiversity) where $\sum_O w_O = 1$, it was calculated how high each of the land-use types k scores in each cell n in terms of the overall weighted benefit b :

$$b_{n,k} = \bar{b}_{n,k}^C \times w_C + \bar{b}_{n,k}^P \times w_P + \bar{b}_{n,k}^B \times w_B$$

Where $\bar{b}_{n,k}^O$ is the normalised benefit to objective O . **Then, for each cell n the LU type k that yields the maximum weighted benefit is chosen:**

$$k_n = \operatorname{argmax}_k b_{n,k_n}$$

$k_1, k_2, k_3, \dots, k_M$ together define a scenario which maximizes the overall weighted benefit $\sum_n b_{n,k_n}$. For each of the created scenarios, the benefits for carbon sequestration, production and biodiversity were calculated, and the pareto efficient scenarios were conserved.”

Edited plot in Line 873 (Line 889):

And revised plot description in Lines 874-878 (Lines 890 – 896):

Figure 1 Overview of the steps in the methodology. ① Conversions from the current land-use are possible to four land-use categories ② The potential benefits of conversions to those four land-use categories are calculated against three objectives: GHG emissions, agricultural/forestry production and biodiversity. ③ The potential benefits are weighted using 231 weighting combinations ④ and summed up to the resulting weighted overall benefit. ⑤ The highest weighted overall benefit and the corresponding optimal land-use conversion is identified.

Manuscript ID: COMMSENV-25-0803B

Oxford, 23/07/2025

Dear Editors,

Thank you for the opportunity to submit the final revised version of our manuscript. We're grateful to the reviewers for their feedback and are pleased that the revisions have been well received. Below, we provide our point-by-point responses to the final comments. All requested changes have been addressed in the manuscript.

Reviewer #1

Comment:

"The author has provided a thorough and clear response to the issues raised, and has made specific revisions to the manuscript, including additions to the analysis and discussion of the results. I think my concerns have been fully addressed, including the justification of the step size and the limitations regarding land conversion feasibility. The additional figures have also improved the readability, and the revised version represents a significant improvement over the original submission. However, I recommend that the author carefully review the use of certain terms in the final manuscript, such as distinguishing between 'carbon storage' and 'carbon sequestration'."

Response:

Thank you for your kind feedback. As suggested, we carefully reviewed the manuscript to ensure that "carbon storage" and "carbon sequestration" are used consistently and accurately.

Reviewer #2

Comment:

"The authors have done a sufficient amount of revisions in response to my previous comments and I appreciate their efforts on improving the ms. I find all the responses satisfying and do not have more comments."

Response:

We appreciate your supportive response and are glad to hear that the revisions addressed your concerns.

Reviewer #3

Comment 1:

"Abbreviations on first use – e.g. 'UK' is introduced early but 'GB' appears later; confirm both are defined on first use."

Response:

Thanks for pointing this out. We've now ensured that all abbreviations, including "UK" and "GB", are defined the first time they appear.

Comment 2:

"Units and symbols – Ensure a non breaking space between number and unit (e.g. '500 m', '25 m raster'), and italicise variables in equations."

Response:

We've gone through the manuscript to add non-breaking spaces between numbers and units instead of normal spaces, and we've italicised all variables in the equations to meet the journals formatting standards.

Comment 3:

"Capitalisation of headings – The journal's style guide may prefer either sentence case or title case; align section and subsection headings accordingly."

Response:

We've edited the heading formatting across the manuscript to match the journal's preferred style.

We hope these final revisions meet your expectations. Many thanks again to the reviewers and editors for their helpful input throughout the process.

Best regards,

Sarah Gall

University of Oxford

sarah.gall@spc.ox.ac.uk

(On behalf of all co-authors)